# Adaptive Thinking: Large Language Models Know When to Think in Latent Space

**Pingzhi Li**[1,2†]   **Bairu Hou**[1]   **Yun Zhu**[1†]   **Yihao Feng**[1]   **Ke Ye**[1†]   **Tao Lei**[1]
**Zhifeng Chen**[1]   **Tianlong Chen**[2]   **Xianzhi Du**[1]

[1]Apple      [2]The University of North Carolina at Chapel Hill

[†]Work done at Apple

## Abstract

Recent advances in large language models (LLMs) test-time computing have introduced the capability to perform intermediate chain-of-thought (CoT) reasoning (thinking) before generating answers. While increasing the thinking budget yields smooth performance improvements at inference time, the relationship between LLM capability, query complexity, and optimal budget allocation remains poorly understood for achieving compute-optimal inference. To address this challenge, we utilize *self-consistency*, the agreement among multiple reasoning paths, as a proxy for thinking necessity. We first identify that lower self-consistency indicates when queries require extended thinking to reach correct answers. Building on this insight, we introduce Sonata (**S**elf-**Con**sistency-Guided **A**dapter for **T**hinking **A**llocation), a lightweight approach that adaptively allocates thinking budgets to optimize the performance-efficiency tradeoff. Sonata includes an adapter trained offline on a calibration dataset to predict self-consistency directly from the last layer hidden representations during the query prefilling stage. This prediction then guides on-the-fly budget allocation before thinking. The adapter is general, transferable across diverse tasks once trained, and introduces $< 1‰$ computational overhead during inference. Notably, Sonata is compatible with existing CoT compression methods, enabling further efficiency gains when managing thinking budgets across queries. Extensive experiments on multiple models (Qwen3-8B, Qwen3-32B, GPT-OSS-120B, Qwen3-235B-A22B) and benchmarks (AIME25, GSM8K, MATH500, GPQA, LiveCodeBench) demonstrate that Sonata achieves $20\%$ to $60\%$ reduction in thinking tokens while maintaining the same accuracy, or up to $2\%$ improvement in accuracy with the same token cost.

## 1 Introduction

The ability to perform extended reasoning at inference time has emerged as a transformative capability for large language models (LLMs), enabling them to tackle complex problems through chain-of-thought (CoT) reasoning (Wei et al., 2022; Kojima et al., 2022). Recent advances in test-time compute scaling have demonstrated that allowing LLMs to *"think"* before answering, generating intermediate CoT reasoning tokens, can yield significant performance improvements on challenging tasks (Snell et al., 2024; Wu et al., 2025a; Yang et al., 2025a; DeepSeek-AI et al., 2025; Gemini Team et al., 2025). This thinking capability enables these LLMs to explore diverse reasoning paths, reflect on their decisions, refine solutions, and rigorously verify correctness, during inference time (DeepSeek-AI et al., 2025; OpenAI, 2024).

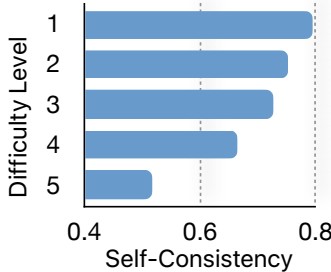

Figure 1: Average *self-consistency* across various difficulty levels, derived from Qwen3-8B model on the MATH-500 task.

However, adaptively determining the optimal thinking budget for each query remains a critical challenge, as excessive thinking wastes computational resources on simple queries and may even hurt performance (Li et al., 2025c; Hassid et al., 2025; Wu et al., 2025b; Hou et al., 2025), while in-

sufficient thinking leads to errors on complex ones (Snell et al., 2024; Wu et al., 2025a). The core problem lies in identifying how much thinking a specific query really requires before generating the response tokens. Existing approaches either rely on superficial proxies like entropy (Xia et al., 2024; Zhang et al., 2025a), which fail to capture the intrinsic reasoning difficulty, or require costly online computation (Han et al., 2025) or sample-specific calibration, limiting their practical applicability.

*Can we enable LLMs to adaptively allocate thinking budget given queries, optimizing the thinking performance-efficiency trade-off at test time?* In this work, we utilize *self-consistency* (Wang et al., 2023; Chen et al., 2023), *i.e.* the agreement among multiple reasoning paths when sampling from the LLM given the same query, as a principled proxy for thinking necessity. As shown in Figure 1, queries of higher difficulty levels demonstrate lower self-consistency scores. Self-consistency directly measures the model's confidence in solving a problem, as queries with high self-consistency (where multiple reasoning attempts converge to the same answer) typically require minimal thinking, while those with low self-consistency benefit from extended CoT reasoning. We further analyze the hidden representations of various queries in thinking LLMs and observe that they are highly distinguishable in the latent space. Building on this insight, we introduce `Sonata` (**S**elf-**Con**sistency-Guided **A**dapter for **T**hinking **A**llocation), a lightweight approach that first learns to predict self-consistency directly from query hidden representations in the last layer. During inference, `Sonata` adapter takes the query's last layer hidden representations as input during the prefilling stage and adaptively allocates thinking budgets before decoding. This adapter, trained offline on a calibration dataset, introduces $< 1\permil$ computational overhead and is generalizable across tasks without task-specific fine-tuning. Moreover, `Sonata` is compatible to LLMs trained with existing CoT compression techniques (Zhang et al., 2025b; Hou et al., 2025; Lu et al., 2025), enabling further efficiency gains while maintaining performance.

Our contributions and findings are summarized as follows: (i) **Self-consistency as a reasoning indicator:** We utilize self-consistency as an effective proxy for evaluating LLMs' reasoning capabilities, revealing that prompts exhibiting different self-consistency levels are highly distinguishable in the latent space; (ii) **Adaptive reasoning with `Sonata`:** We introduce `Sonata`, a lightweight adapter with negligible cost that adaptively determines both *when* to conduct reasoning and *how much* reasoning budget to allocate. (iii) **Superior performance-efficiency tradeoff:** Extensive experiments across models of various scales (Qwen3-8B, Qwen3-32B, GPT-OSS-120B, Qwen3-235B-A22B) and tasks of various difficulties (AIME25, GSM8K, MATH500, GPQA, LiveCodeBench) validate that `Sonata` reduces average token consumption by up to $60\%$ while maintaining task performance.

## 2 RELATED WORKS

**Token Efficiency for Thinking LLMs.** Recent research has attempted to attack the efficiency challenge of LLM thinking by reducing the number of tokens spent. One line of work focuses on post-training. Hassid et al. (2025) observes that shorter reasoning chains are often more accurate and proposes an early-exit inference strategy. Similarly, Jiang et al. (2025) employs a verification model to decide when to terminate the reasoning process. Other methods intervene more directly during generation. Li et al. (2025a) inserts a reasoning terminator token early based on attention analysis, while Qiao et al. (2025) uses a confidence-guided approach to suppress redundant reflection steps. Another category of methods uses reinforcement learning (RL) to encourage brevity. Hou et al. (2025) uses RL with a token limit to prune long chains of thought, and Yi et al. (2025) defines a Sample Optimal Length to guide the model toward more efficient outputs. Zhang et al. (2025b) introduces a length-regularized RL method, and Li et al. (2025c) trains a model to pre-estimate its own token budget. Some approaches refine the training data itself; for instance, Lu et al. (2025) uses a search algorithm to discover shorter, more effective reasoning paths for distillation. Notably, our work is compatible with these compression and pruning techniques. By adaptively allocating a thinking budget *before* the reasoning process begins, `Sonata` can be combined with these methods to further optimize the performance-efficiency tradeoff across queries.

**Self-Consistency in LLMs.** Self-consistency has been proposed as a decoding strategy that improves CoT reasoning by sampling multiple diverse reasoning paths and selecting the answer that appears most frequently via majority vote (Wang et al., 2023). While effective, this approach incurs significant computational costs and is primarily applicable to tasks with easily extractable, closed-form answers. To overcome these limitations, subsequent research has focused on enhancing SC's

Figure 2: Correlation between self-consistency and thinking improvement across five difficulty levels on MATH-500 using Qwen3-8B. Each point denotes an individual query, with self-consistency computed from $N = 32$ samples in non-thinking mode (x-axis) and accuracy improvement from enabling thinking averaged over 3 runs (y-axis).

efficiency and applicability. Chen et al. (2023) extends the method to free-form generation tasks by using the LLM itself to identify the most consistent response among multiple candidates, removing the need for answer extraction. Other works aim to reduce the high sampling cost. Wan et al. (2025) introduces an early stopping mechanism by evaluating the quality of the intermediate reasoning paths, not just the final answers. Wang et al. (2025) propose to first use the LLM to assess a query's difficulty, then allocate a proportional sampling number, which saves resources on simpler problems. In this work, we utilize the insight that self-consistency can serve as a proxy of the need for extended CoT reasoning, enabling us to allocate the thinking budget adaptively.

**Reasoning in Latent Space.** Recent studies show that LLMs implicitly perform latent reasoning within their hidden computations (Yang et al., 2025b; Shalev et al., 2024b; Lindsey et al., 2025; Tack et al., 2025). This line of research investigates how LLMs process multi-hop queries by maintaining distributions over potential intermediate answers in hidden states, a mechanism that persists even without sufficient knowledge for correct answers (Shalev et al., 2024a). Beyond discrete tokens, recent work trains models to reason directly in continuous latent space by recirculating hidden states as inputs, enabling efficient patterns like breadth-first search (Hao et al., 2024). This latent reasoning is controllable by identifying representations of thought patterns (*e.g.*, execution, reflection), targeted interventions can steer reasoning processes to improve accuracy and efficiency (Chen et al., 2025). Alternative approaches construct "soft" concept tokens from probability-weighted embeddings to implicitly explore multiple reasoning trajectories (Zhang et al., 2025c), or enhance pretraining by integrating continuous concepts extracted via sparse autoencoders into hidden states (Tack et al., 2025). These findings collectively suggest the underlying connection between reasoning capabilities and latent representations, motivating our use of hidden states for adaptive thinking.

## 3 PRELIMINARY

In this section, we provide the foundation for adaptive thinking allocation in LLMs. We first demonstrate that self-consistency serves as a reliable indicator for when models need extended chain-of-thought reasoning in Section 3.1. We then show that self-consistency patterns are distinguishable in the latent space, enabling efficient prediction from hidden representations in Section 3.2.

### 3.1 SELF-CONSISTENCY INDICATING WHEN TO THINK

Self-consistency has initially emerged as a powerful decoding strategy that enhances CoT reasoning in large language models by leveraging the intuition that complex reasoning problems often admit multiple valid reasoning paths leading to the same correct answer (Wang et al., 2023; Chen et al., 2023). When an LLM generates multiple CoT chains for the same query, the consistency among their final answers serves as a strong indicator of the model's confidence and reasoning capability on that particular query. Inspired by these existing works, we hypothesize that for reasoning models,

*low **self-consistency** indicates the need for extended CoT reasoning.*

Formally, we define self-consistency as the ratio of correct samples among multiple repeated sampling. Given a query $q$ and an LLM $\mathcal{M}$, we sample $N$ independent answers $\{a_1, a_2, ..., a_N\}$ [1]. The self-consistency score is computed as $\text{SC}(q) = \frac{1}{N} \sum_{i=1}^{N} \mathbb{I}[a_i = a^*]$, where $\mathbb{I}[\cdot]$ is the indicator function and $a^*$ denotes the correct answer [2].

---

[1] We used $N = 32$ for all experiments in our work.

[2] We employ a verifier to determine correctness rather than relying solely on majority voting, which allows us to accurately assess self-consistency for calibration.

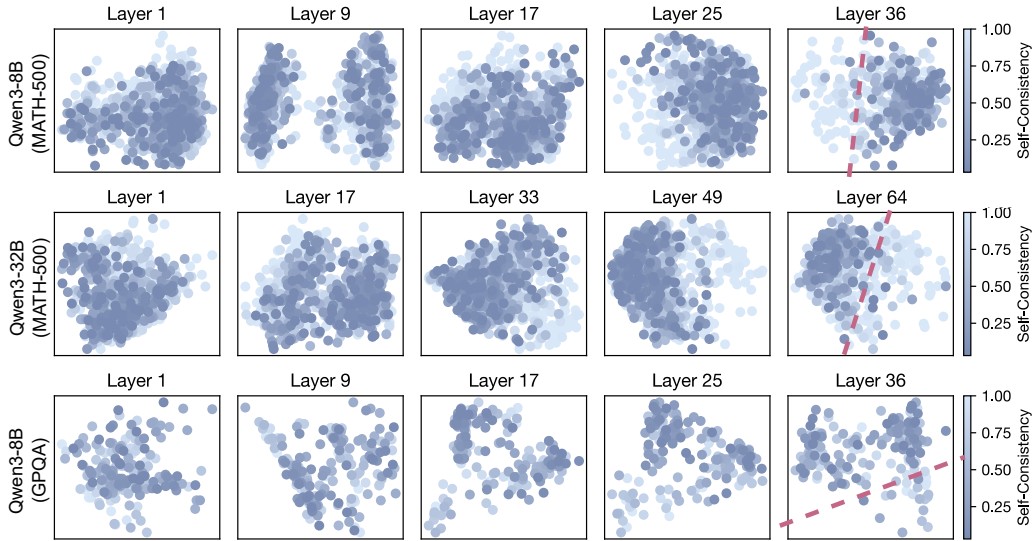

Figure 3: PCA visualization of query hidden representations across different transformer layers, colored by self-consistency scores, evaluated on both MATH-500 (math reasoning) and GPQA (scientific reasoning) benchmarks. Self-consistency patterns become increasingly distinguishable in deeper layers, with the last layers (*i.e.* 36, 64) showing the most pronounced separation. High self-consistency queries (**dark**) form tight clusters while low self-consistency queries (**light**) are more dispersed, demonstrating that self-consistency signals are learnable from latent representations across diverse reasoning domains.

To investigate the relationship between self-consistency and the necessity for extended thinking, we conduct experiments measuring the performance gain from thinking *vs.* non-thinking modes. Specifically, for each query $q$ in our calibration set, we compute: ❶ the self-consistency score $\text{SC}(q)$ when the model operates without thinking, by enforcing the thinking terminate token `</think>` right after starting token `<think>`, following Yang et al. (2025a); DeepSeek-AI et al. (2025), and ❷ the accuracy improvement $\Delta_{\text{think}}(q) = \text{Acc}_{\text{think}}(q) - \text{Acc}_{\text{non-think}}(q)$, where $\text{Acc}_{\text{think}}$ and $\text{Acc}_{\text{non-think}}$ represent the accuracy with and without chain-of-thought reasoning, respectively. As illustrated in Figure 2, we observe a strong negative correlation between self-consistency in non-thinking mode and the performance gains from thinking. Each point is an individual query from our calibration dataset. Queries with low self-consistency exhibit significant improvements when thinking is enabled, while queries with high self-consistency show minimal improvements. The cluster of points in the lower-left corner represents intrinsically difficult problems where both self-consistency and thinking improvements are low, indicating queries that remain challenging even with extended reasoning. These empirical results validate our hypothesis that self-consistency serves as a principled indicator for adaptive thinking budget allocation.

## 3.2 Self-Consistency Patterns are Distinguishable in Latent Space

While self-consistency provides a reliable signal for thinking necessity, computing it requires expensive repeated sampling that defeats the purpose of efficient inference. This raises a critical question: *can we predict self-consistency directly from the model's internal representations without explicit sampling?* We present two key observations that enable efficient self-consistency prediction.

**Observation 1: Self-consistency patterns are highly distinguishable in latent representations.** We analyze the hidden states of queries with varying self-consistency levels by extracting the last token's representation from the final transformer layer. Specifically, given a query $q$ with the chat template, we obtain the hidden state $\mathbf{H} \in \mathbb{R}^d$ from the last position before any decoding begins. We apply Principal Component Analysis (PCA) to project $\mathbf{H}$ onto a two-dimensional space. As shown in Figure 3, queries naturally cluster according to their self-consistency levels in this projected space. High self-consistency queries (darker blue) naturally form tight clusters, indicating similar reasoning patterns, while low self-consistency queries (lighter blue) are more dispersed.

**Observation 2: Deeper layers exhibit stronger self-consistency separability.** We further investigate how self-consistency patterns evolve across different transformer layers. Let $\mathbf{H}^{(l)}$ denote the

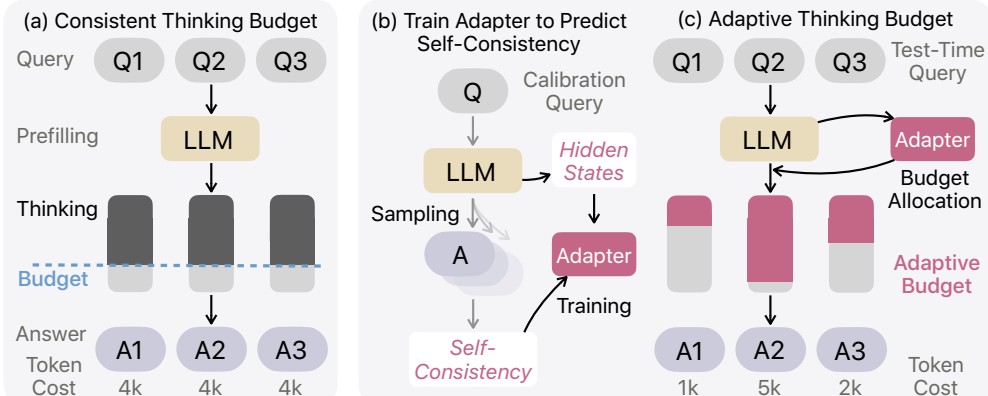

Figure 4: Overview of `Sonata`. (a) Conventional approaches (Yang et al., 2025a) where all queries receive the same fixed thinking budget (*e.g.* 4k tokens each) during thinking, regardless of query complexity, resulting in suboptimal token allocation. (b) Offline training phase where the MLP adapter learns to predict *self-consistency* from last-layer hidden states. For each calibration query **Q**, the LLM generates multiple responses, and self-consistency is computed as the accuracy among all sampled answers **A**. The adapter is trained to map the last hidden states to these self-consistency scores. (c) At inference time, `Sonata` employs the trained adapter to predict self-consistency from query hidden states during prefilling and adaptively allocates lower thinking budgets to higher self-consistency queries (*e.g.* **Q1** for 1k), reducing overall token cost while maintaining accuracy.

---

**Algorithm 1** Offline Self-Consistency Adapter Training

---

**Require:** Calibration dataset $\mathcal{D}_{\text{cal}} = \{q_i\}_{i=1}^K$, LLM $\mathcal{M}$, sampling size $N$
**Ensure:** Trained adapter $f_\theta$
1: Initialize $\mathcal{S} \leftarrow \emptyset$
2: **for** $i = 1 \rightarrow K$ **do**
3:     Sample $\mathcal{A}_i \leftarrow \{a_j \sim P_\mathcal{M}(\cdot|q_i, \text{non-thinking})\}_{j=1}^N$
4:     Compute $\text{SC}_i \leftarrow \frac{1}{N} \sum_{j=1}^N \mathbb{I}[a_j = a_i^*]$
5:     Extract $\mathbf{h}_i \leftarrow \text{LLM}_L(q_i)$ {Last layer, last token}
6:     $\mathcal{S} \leftarrow \mathcal{S} \cup \{(\mathbf{h}_i, \text{SC}_i)\}$
7: **end for**
8: $f_\theta \leftarrow \text{Train}(\mathcal{S}, \mathcal{L}_{\text{MSE}})$ {MSE loss}
9: **return** $f_\theta$

---

hidden representation at layer $l$. Figure 3 demonstrates that self-consistency becomes increasingly distinguishable in deeper layers, with the final layer showing the most pronounced separation. This aligns with literature that deeper layers encode more abstract, conceptual, and reasoning-related knowledge, while shallow layers primarily capture low-level linguistic features (Rogers et al., 2020; Jin et al., 2025). The strong self-consistency signals in final-layer representations enable our efficient on-the-fly adapter-based prediction approach, presented in Section 4.

## 4 METHODOLOGY

In this section, we present the `Sonata` framework for adaptive thinking budget. We introduce a lightweight adapter that learns to predict self-consistency from query representations in Section 4.1. Section 4.2 describes how this adapter enables on-the-fly thinking budget allocation during inference with negligible computational overhead, achieving optimal performance-efficiency trade-offs.

### 4.1 TRAINING ADAPTER TO PREDICT SELF-CONSISTENCY

We train a lightweight adapter to predict self-consistency directly from query representations, eliminating the need for expensive sampling during inference. Given a calibration dataset $\mathcal{D}_{\text{cal}} = \{q_1, q_2, ..., q_K\}$, we first collect self-consistency labels by sampling $N$ answers for each query $q_k$ in non-thinking mode, by enforcing `</think>` immediately after `<think>`. The self-consistency score is computed as $\text{SC}(q_k) = \frac{1}{N} \sum_{i=1}^N \mathbb{I}[a_k^{(i)} = a_k^*]$, where $a_k^*$ is the ground-truth answer.

---

**Algorithm 2** Online Adaptive Thinking Decision

---

**Require:** Test query $q$, trained adapter $f_\theta$, LLM $\mathcal{M}$, threshold $\tau_0$
**Ensure:** Response $r$ with adaptive thinking
 1: Extract $\mathbf{h} \leftarrow \text{LLM}_L(q)$ {Prefilling stage: last layer, last token}
 2: Predict $\hat{s} \leftarrow f_\theta(\mathbf{h})$ {Self-consistency prediction}
 3: **if** $\hat{s} > \tau_0$ **then**
 4:     $r \leftarrow \text{Generate}(\mathcal{M}, q, \text{thinking} = \text{False})$ {Direct answer, no thinking}
 5: **else**
 6:     $r \leftarrow \text{Generate}(\mathcal{M}, q, \text{thinking} = \text{True})$ {Generate with thinking}
 7: **end if**
 8: **return** $r$

---

As shown in Figure 4 (b), for each query with the chat template, right before decoding, we extract the last token's hidden representation $\mathbf{h}_k$ from the final transformer layer during prefilling. We then train a two-layer MLP adapter $f_\theta$ to map these representations to self-consistency scores, followed by a `sigmoid` mapping function. Algorithm 1 formally demonstrates the training procedure. Once trained offline on the calibration dataset, the `Sonata` adapter is generalizable across queries of diverse tasks without additional fine-tuning, introducing negligible computational overhead during inference as it requires only a single forward pass through the lightweight MLP.

### 4.2 ON-THE-FLY THINKING BUDGET ALLOCATION WITH ADAPTER

At test time, our trained adapter enables adaptive thinking budget allocation with negligible computational overhead. The adapter is model-specific—trained for each LLM architecture, while task-agnostic, generalizing across diverse downstream queries without retraining. As illustrated in Figure 4(c), `Sonata` dynamically determines both whether to engage thinking and how much budget to allocate based on the query's hidden representation.

Specifically, given a test-time query $q$, we extract its hidden representation $\mathbf{h} = \text{LLM}_L(q)$ during the prefilling stage, just before decoding begins. The adapter then predicts the self-consistency score $\hat{s} = f_\theta(\mathbf{h})$, which serves as our confidence indicator. With this prediction, we determine whether to think for a given query. We compare $\hat{s}$ against a predefined threshold $\tau_0$ [3]. If $\hat{s} > \tau_0$, the model proceeds without thinking (directly generating the answer), as high predicted self-consistency indicates the query is straightforward. Otherwise, thinking is conducted with the model's default thinking process. Algorithm 2 formally presents the online inference procedure.

The entire allocation process requires only a single forward pass through the lightweight MLP adapter, introducing virtually zero latency compared to the LLM's inference time. Since the adapter operates on already-computed hidden states from prefilling, no additional LLM forward passes are needed. This enables `Sonata` to adaptively decide whether to think based on real-time query complexity, unlike the fixed allocation approach in Figure 4(a), significantly reducing average token consumption while maintaining performance.

## 5 EMPIRICAL EVALUATION

In this section, we present comprehensive experiments evaluating `Sonata`'s effectiveness in adaptive thinking budget allocation. In Section 5.1, we present our main experimental results across four thinking-capable models of varying scales (8B to 235B parameters) on four challenging reasoning benchmarks, demonstrating its effectiveness and efficiency. Section 5.2 provides detailed ablation and extended studies examining the impact of different proxy metrics, adapter architectures, threshold configurations, and the computational overhead during inference.

### 5.1 MAIN RESULTS

**Experimental Setup.** We evaluate `Sonata` on five challenging reasoning benchmarks: AIME25, GSM8K, MATH500, LiveCodeBench, and GPQA, covering mathematical, code generation, and general reasoning tasks and across diverse difficulties. We conduct experiments on four thinking

---

[3]We find that $\tau_0 = 0.3$ generally works well, and thus we set it to 0.3 for all experiments in this work.

Table 1: Comparison results on the AIME25, MATH-500, GSM8K, LiveCodeBench (LCB) and GPQA across four models with thinking capability. We use $\texttt{temperature} = 0.6$, $\texttt{top\_p} = 0.95$ for decoding. We report the average performance of three repeated trials for each run. Accuracy (Acc.) comparable to or higher than the vanilla baseline model are underlined, and the lowest thinking token counts (#Tokens) among those with underlined accuracy are marked in **bold**.

| Methods | AIME25 Acc. (↑) | #Tokens (↓) | MATH-500 Acc. (↑) | #Tokens (↓) | GSM8K Acc. (↑) | #Tokens (↓) | LCB Acc. (↑) | #Tokens (↓) | GPQA Acc. (↑) | #Tokens (↓) | Average Acc. (↑) | #Tokens (↓) |
|---|---|---|---|---|---|---|---|---|---|---|---|---|
| Qwen3-8B | 60.0 | 16995 | 97.6 | 4900 | 95.2 | 1994 | 57.8 | 14421 | 60.1 | 7458 | 74.1 | 9154 |
| w. Const. Budget | 30.0 | 4096 (24%) | 93.2 | 4096 (84%) | 95.0 | 4096 (205%) | 49.3 | 4096 (28%) | 57.1 | 4096 (55%) | 64.9 | 4096 (45%) |
| w. Self-Judge | 60.0 | 17019 (100%) | 96.0 | 4315 (88%) | 92.7 | 1076 (54%) | 57.1 | 13496 (93%) | 57.6 | 5913 (79%) | 72.7 | 8364 (91%) |
| w. Sonata | 63.3 | **16449 (97%)** | 97.4 | **3694 (75%)** | 95.6 | **890 (45%)** | 58.2 | **13054 (90%)** | 62.0 | **3590 (48%)** | 75.3 | **7535 (82%)** |
| Qwen3-32B | 70.0 | 14971 | 97.6 | 3670 | 94.8 | 1654 | 63.8 | 13729 | 63.6 | 5329 | 78.0 | 7853 |
| w. Const. Budget | 56.7 | 4096 (27%) | 95.8 | 4096 (112%) | 95.3 | 4096 (248%) | 47.0 | 4096 (30%) | 64.1 | 4096 (77%) | 69.8 | 4096 (52%) |
| w. Self-Judge | 73.3 | 15625 (104%) | 96.2 | 3107 (85%) | 93.1 | 666 (40%) | 59.3 | 13360 (97%) | 54.5 | 3807 (71%) | 75.3 | 7313 (93%) |
| w. Sonata | 70.0 | **14890 (100%)** | 98.0 | **2583 (70%)** | 94.4 | **728 (44%)** | 63.4 | **13600 (99%)** | 63.1 | **3568 (70%)** | 77.8 | **7074 (90%)** |
| GPT-OSS-120B-High | 86.7 | 14390 | 98.2 | 2331 | 85.8 | 494 | - | - | 75.8 | 9617 | 86.6 | 6708 |
| w. Const. Budget | 73.3 | 4096 (28%) | 98.0 | 4096 (176%) | 86.4 | 4096 (829%) | - | - | 69.2 | 4096 (43%) | 81.7 | 4096 (61%) |
| w. Self-Judge | 83.3 | 14045 (98%) | 94.4 | 890 (38%) | 87.0 | 105 (21%) | - | - | 64.1 | 5894 (61%) | 82.2 | 5234 (78%) |
| w. Sonata | 86.7 | **13817 (96%)** | 98.0 | **1683 (72%)** | 86.7 | **385 (78%)** | - | - | 70.2 | **8008 (83%)** | 85.4 | **5973 (89%)** |
| Qwen3-235B-A22B | 70.0 | 13831 | 97.6 | 4371 | 94.2 | 2261 | - | - | 69.2 | 7049 | 82.8 | 6878 |
| w. Const. Budget | 43.3 | 4096 (30%) | 94.8 | 4096 (94%) | 95.7 | 4096 (181%) | - | - | 60.1 | 4096 (58%) | 73.5 | 4096 (60%) |
| w. Self-Judge | 73.3 | 13951 (101%) | 98.0 | 4012 (92%) | 93.3 | 1037 (46%) | - | - | 68.2 | 6658 (94%) | 83.2 | 6415 (93%) |
| w. Sonata | 73.3 | **13890 (100%)** | 98.0 | **2984 (68%)** | 94.0 | **998 (44%)** | - | - | 70.7 | **4919 (70%)** | 84.0 | **5698 (83%)** |

models of varying scales: Qwen3-8B, Qwen3-32B, GPT-OSS-120B, and Qwen3-235B-A22B. We evaluate Qwen3-8B and Qwen3-32B in BF16, GPT-OSS-120B in MXFP4, and Qwen3-235B-A22B in FP8. For calibration dataset construction, we randomly sample 1000 problems from the Open-MathReasoning [4] dataset, specifically selecting difficulty level 6 and 7 problems to ensure sufficient complexity and diversity for training the adapter. During calibration, we use $N = 32$ samples per query to compute ground-truth self-consistency scores in non-thinking mode. For all inference experiments, we employ sampling parameters with $\texttt{top\_p} = 0.95$ and $\texttt{temperature} = 0.6$. For each evaluation runs, we conduct four repeated trials with random seeds of $\{233, 234, 235\}$ for reproducibility. We report pass@1 accuracy for all tasks. All of our experiments are conducted on NVIDIA B200 GPU servers. We compare Sonata with two baselines: (1) *constant thinking budget control* (Yang et al., 2025a), by predefining the thinking budget and inserting a thinking termination token when reaching the budget during decoding; and (2) *self-judged thinking budget*, by first asking the LLM to decide a thinking budget before decoding the response with or without thinking enabled.

**Competitive Efficiency and Accuracy.** As shown in Table 1, Sonata demonstrates substantial efficiency improvements across all evaluated models and benchmarks while maintaining or improving accuracy. Several conclusions can be drawn: ❶ Sonata achieves the best efficiency-performance trade-off on nearly all benchmarks and all models, consistently outperforming both baselines in terms of token reduction while maintaining comparable or superior accuracy. For instance, on Qwen3-8B, Sonata improves accuracy by $1.4\%$ while reducing tokens by $21\%$ compared to the vanilla model. ❷ The efficiency gains are particularly pronounced on simpler tasks such as GSM8K and MATH-500 across all models. On GSM8K, Sonata reduces token usage by $55\%$–$56\%$ for smaller models (Qwen3-8B and Qwen3-32B) while maintaining accuracy, likely because simpler tasks are more prone to overthinking and wasted tokens, where Sonata provides effective guidance to mitigate this problem. ❸ Sonata shows greater improvements on weaker models, such as comparing Qwen3-8B's $79.6\%$ average accuracy with $21\%$ token savings against Qwen3-235B-A22B's $84.0\%$ accuracy with $17\%$ token savings. This suggests that weaker models are more susceptible to overthinking, and Sonata effectively guides them toward compute-optimal thinking allocation. ❹ The self-judge baseline shows sometimes decent but inconsistent performance. While it occasionally maintains accuracy and reduces tokens, especially on larger models (e.g., Qwen3-235B-A22B achieves competitive performance on AIME25 and MATH-500 with only $1\%$ accuracy drop on GSM8K and GPQA), it generally underperforms Sonata. This pattern indicates that larger models potentially possess better self-assessment capabilities for determining thinking necessity on given queries, though our learned adapter approach remains more reliable overall.

**Generalization Across Tasks.** A key strength of Sonata is its ability to generalize beyond the calibration domain. Despite being trained exclusively on mathematical problems from the OpenMathReasoning dataset, the adapter demonstrates strong transfer to both the GPQA benchmark, which requires general scientific reasoning across physics, chemistry, and biology, and Live-CodeBench, which evaluates code generation capabilities. As shown in Table 1, Sonata achieves

---

[4]https://huggingface.co/datasets/nvidia/OpenMathReasoning

particularly impressive results on these out-of-domain tasks. For example, it improves the accuracy of GPQA by 1.9% for Qwen3-8B while reducing tokens by 52%, the highest accuracy improvement across all tasks. On LiveCodeBench, `Sonata` achieves 58.2% pass@1 for Qwen3-8B and 63.4% for Qwen3-32B, while reducing tokens by up to 10% compared to vanilla models. Interestingly, we observe that the token savings on GPQA (from 17% to 52% across models) are comparable to those on mathematical tasks, despite the domain varying. This pattern holds consistently across all model scales in our evaluation. The consistent performance across domains suggests that self-consistency patterns capture fundamental aspects of reasoning difficulty that transcend specific subject matters. This cross-domain generalization validates that the reasoning difficulty indicators learned through self-consistency reflect general properties of query complexity and model capability rather than domain-specific characteristics, making `Sonata` practical for real-world deployment where queries often span multiple knowledge domains.

## 5.2 ABLATION AND EXTENDED RESULTS

**End-to-End Memory and Latency.** Table 2 presents comprehensive end-to-end efficiency metrics on the MATH-500 benchmark. The lightweight adapter introduces negligible memory overhead, with less than 1% increase even for the smallest model. The detailed analysis can be found in Appendix B. More significantly, `Sonata` achieves substantial latency reductions ranging from 27% (*i.e.*, Qwen3-8B) to 36%

Table 2: End-to-end inference efficiency comparison. Peak memory usage and latency are measured on NVIDIA B200 GPUs with batch size 1. Results are evaluated on MATH-500 and averaged across all queries. Peak memory is tested via HuggingFace Inference, while latency and throughput are tested via vLLM. Qwen3-235B-A22B is evaluated on two B200 GPUs, while other 3 models are evaluated on one.

| Model | Memory (GB) (↓) | | Latency (s) (↓) | | Throughput (tokens/s) (↑) | |
|---|---|---|---|---|---|---|
| | Base | w. Sonata | Base | w. Sonata | Base | w. Sonata |
| Qwen3-8B | 17 | **16** | 32.1 | **23.5** | 153 | **157** |
| Qwen3-32B | 62 | **61** | 32.2 | **21.5** | 114 | **120** |
| GPT-OSS-120B | 63 | **62** | 13.1 | **9.3** | 215 | **245** |
| Qwen3-235B-A22B | 238 | **237** | 81.0 | **51.5** | 54 | **58** |

(*i.e.*, Qwen3-235B-A22B) by eliminating unnecessary thinking tokens. Interestingly, larger models benefit more from adaptive allocation. Qwen3-235B-A22B shows the greatest latency reduction (29.5 seconds saved per query), as the cost of generating thinking tokens scales with model size. Overall, `Sonata` demonstrates consistent computational cost in terms of memory and latency, particularly for larger models where thinking token costs more inference time.

**Improved Pareto Frontier.** Figure 5 demonstrates that `Sonata` achieves a superior accuracy-efficiency Pareto frontier compared to the constant budget baseline across model scales. By adjusting the adapter's self-consistency threshold $\tau_0$ from 0 to 1, we enable smooth trade-offs between accuracy and token consumption. The constant budget approach forces a uniform thinking allocation across all queries, resulting in suboptimal performance, which either sacrifices accuracy on complex problems or wastes computation on simple ones. In contrast, `Sonata` provides fine-grained, query-adaptive control.

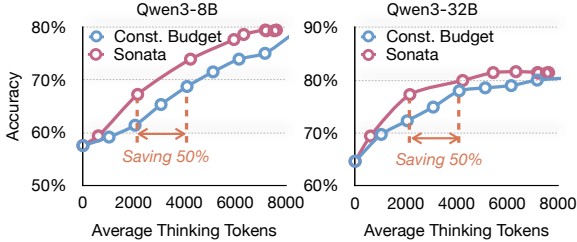

Figure 5: Accuracy-efficiency Pareto frontiers comparing `Sonata` against constant budget baseline on Qwen3-8B and Qwen3-32B. By adjusting the self-consistency threshold $\tau_0$, `Sonata` consistently outperforms the fixed budget approach, achieving up to 50% token savings at comparable accuracy levels.

For Qwen3-8B, `Sonata` maintains consistently higher accuracy across all token budgets, with accuracy improvements reaching up to 10% at similar computational costs. This superiority becomes more pronounced with larger models, *i.e.* Qwen3-32B. This superior Pareto frontier demonstrates `Sonata`'s more efficient utilization of thinking tokens.

**Different Proxy Metrics.** Table 3 compares self-consistency against alternative proxy metrics for predicting thinking necessity. ❶ *LM logits entropy* (Fu et al., 2025), computed from the softmax distribution of the last prompt token, performs poorly across all benchmarks. On AIME25, it achieves only 23.3% and 33.3% accuracy for Qwen3-8B and Qwen3-32B respectively. This significant accuracy drop reveals that single-token-level uncertainty fails to capture true reasoning difficulty. ❷ *Attention entropy* (Li et al., 2025b), computed from the attention score in the last layer of the last prompt token, also shows inadequate performance, with 30.0% and 40.0% accuracy on AIME25.

Table 3: Comparison of different proxy metrics for adaptive thinking allocation on four benchmarks. We report accuracy (Acc.) and average thinking tokens (#Tokens) across Qwen3-8B and Qwen3-32B models. Results show that self-consistency (`Sonata`) substantially outperforms two entropy-based metrics, *i.e.* LM logits entropy and Attention entropy, in the accuracy-efficiency tradeoff.

| Methods | AIME25 Acc. (↑) | AIME25 #Tokens (↓) | MATH-500 Acc. (↑) | MATH-500 #Tokens (↓) | GSM8K Acc. (↑) | GSM8K #Tokens (↓) | GPQA Acc. (↑) | GPQA #Tokens (↓) | Average Acc. (↑) | Average #Tokens (↓) |
|---|---|---|---|---|---|---|---|---|---|---|
| Qwen3-8B | 60.0 | 16995 | 97.6 | 4900 | 95.2 | 1994 | 60.1 | 7458 | 78.2 | 7837 |
| *w.* LM Logits Entropy | 23.3 | 2755 (16%) | 93.3 | 4302 (88%) | 93.5 | 1444 (72%) | 55.1 | 5572 (75%) | 66.3 | 3518 (45%) |
| *w.* Attention Entropy | 30.0 | 3124 (18%) | 94.0 | 4185 (85%) | 92.8 | 1389 (70%) | 56.3 | 5891 (79%) | 68.3 | 3647 (47%) |
| *w.* Self-Consistency (`Sonata`) | 63.3 | 16449 (97%) | 97.4 | 3694 (75%) | 95.6 | 890 (45%) | 62.0 | 3590 (48%) | 79.6 | 6156 (79%) |
| Qwen3-32B | 70.0 | 14971 | 97.6 | 3670 | 94.8 | 1654 | 63.6 | 5329 | 81.5 | 6406 |
| *w.* LM Logits Entropy | 33.3 | 2488 (17%) | 94.2 | 3215 (88%) | 92.9 | 1075 (65%) | 57.6 | 4329 (81%) | 69.5 | 2777 (43%) |
| *w.* Attention Entropy | 40.0 | 2915 (19%) | 95.0 | 3088 (84%) | 93.2 | 1188 (72%) | 58.6 | 4542 (85%) | 71.7 | 2933 (46%) |
| *w.* Self-Consistency (`Sonata`) | 70.0 | 14890 (100%) | 98.0 | 2583 (70%) | 94.4 | 728 (44%) | 63.1 | 3568 (70%) | 81.4 | 5442 (85%) |

Both entropy-based methods exhibit a critical flaw that they both underallocate thinking budget to complex problems, *i.e.* AIME25, where extended reasoning is actually essential. ❸ *Self-consistency* (`Sonata`) demonstrates substantially superior accuracy and efficiency by directly measuring the model's ability to consistently solve the problem across multiple attempts, capturing the inherent reasoning difficulty rather than surface-level LM uncertainty. For example, a query might have high entropy of the single next token due to multiple valid phrasings but still be easily solvable, whereas low self-consistency reliably indicates fundamental reasoning challenges. This validates the effectiveness of our self-consistency as the query's reasoning difficulty proxy.

Table 4: Comparison of different adapter architectures for self-consistency prediction. We evaluate linear projection, 2-layer MLP (`Sonata`), and 3-layer MLP across Qwen3-8B and Qwen3-32B models. The 2-layer MLP achieves superior tradeoff between accuracy and efficiency.

| Methods | AIME25 Acc. (↑) | AIME25 #Tokens (↓) | MATH-500 Acc. (↑) | MATH-500 #Tokens (↓) | GSM8K Acc. (↑) | GSM8K #Tokens (↓) | GPQA Acc. (↑) | GPQA #Tokens (↓) | Average Acc. (↑) | Average #Tokens (↓) |
|---|---|---|---|---|---|---|---|---|---|---|
| Qwen3-8B | 60.0 | 16995 | 97.6 | 4900 | 95.2 | 1994 | 60.1 | 7458 | 78.2 | 7837 |
| *w.* Linear | 56.7 | 15895 (94%) | 96.8 | 3871 (79%) | 94.1 | 917 (46%) | 59.6 | 3804 (51%) | 76.8 | 6122 (78%) |
| *w.* 3-Layer MLP | 63.0 | 16321 (96%) | 97.2 | 3759 (77%) | 95.4 | 913 (46%) | 61.6 | 3582 (49%) | 79.3 | 6144 (78%) |
| *w.* 2-Layer MLP (`Sonata`) | 63.3 | 16449 (97%) | 97.4 | 3694 (75%) | 95.6 | 890 (45%) | 62.0 | 3590 (48%) | 79.6 | 6156 (79%) |
| Qwen3-32B | 70.0 | 14971 | 97.6 | 3670 | 94.8 | 1654 | 63.6 | 5329 | 81.5 | 6406 |
| *w.* Linear | 63.3 | 14272 (95%) | 96.8 | 2908 (79%) | 93.6 | 761 (46%) | 61.1 | 3747 (70%) | 78.7 | 5422 (85%) |
| *w.* 3-Layer MLP | 69.7 | 14935 (100%) | 97.8 | 2642 (72%) | 94.3 | 712 (36%) | 63.6 | 3659 (69%) | 81.4 | 5487 (86%) |
| *w.* 2-Layer MLP(`Sonata`) | 70.0 | 14890 (100%) | 98.0 | 2583 (70%) | 94.4 | 728 (44%) | 63.1 | 3568 (70%) | 81.4 | 5442 (85%) |

**Different Adapter Design.** Table 4 evaluates various adapter architectures for self-consistency prediction. We compare the performance between a simple linear projector *vs.* our 2-layer MLP in `Sonata` *vs.* a 3-layer MLP. Experimental results demonstrate the superiority of our MLP for learning self-consistency from the query last-layer representations. Specifically, our 2-layer MLP consistently achieves 79.6% average accuracy on Qwen3-8B and 81.4% on Qwen3-32B, while the linear projector achieves approximately 3% lower despite similar token usage. Moreover, adding a third layer (*i.e.* the 3-layer MLP) provides diminishing returns, with nearly same accuracy and efficiency compared to our 2-layer MLP. This finding aligns with our observation that self-consistency clusters are well-separated in the latent space, as shown in Section 3.2, suggesting that a lightweight non-linear adapter is sufficient to learn the decision boundaries, while the linear projector struggles to capture non-linear relationships between hidden states and self-consistency patterns.

**Calibration Set Size.** To further test the robustness of `Sonata` under resource-constrained scenarios, we evaluate the adapter with reduced calibration datasets of 100 and 200 samples, compared to the original 1000 samples in Table 1. Table 5 presents results on Qwen3-8B across all benchmarks. Remarkably, `Sonata` maintains consistent performance even with only 100 calibration samples, achieving 79.0% average accuracy while reducing tokens by 17%, compared to 79.6% accuracy and 21% token reduction with 1000 samples. The adapter with 200 samples nearly matches the full calibration performance. These results demonstrate that `Sonata` is robust to calibration set size and can be effectively deployed in low-resource scenarios.

Table 5: Ablation study on calibration dataset size. Results on Qwen3-8B across all benchmarks with 1000, 200, and 100 calibration samples. Token percentages are relative to the vanilla model.

| Methods | AIME25 Acc. (↑) | AIME25 #Tokens (↓) | MATH-500 Acc. (↑) | MATH-500 #Tokens (↓) | GSM8K Acc. (↑) | GSM8K #Tokens (↓) | GPQA Acc. (↑) | GPQA #Tokens (↓) | Average Acc. (↑) | Average #Tokens (↓) |
|---|---|---|---|---|---|---|---|---|---|---|
| Qwen3-8B | 60.0 | 16995 | 97.6 | 4900 | 95.2 | 1994 | 60.1 | 7458 | 78.2 | 7837 |
| *w.* Const. Budget | 30.0 | 4096 (24%) | 93.2 | 4096 (84%) | 95.0 | 4096 (205%) | 57.1 | 4096 (55%) | 68.8 | 4096 (52%) |
| *w.* Self-Judge | 60.0 | 17019 (100%) | 96.0 | 4315 (88%) | 92.7 | 1076 (54%) | 57.6 | 5913 (79%) | 76.6 | 7080 (90%) |
| *w.* `Sonata` (1k samples) | 63.3 | 16449 (97%) | 97.4 | 3694 (75%) | 95.6 | 890 (45%) | 62.0 | 3590 (48%) | 79.6 | 6156 (79%) |
| *w.* `Sonata` (200 samples) | 60.0 | 16990 (100%) | 97.0 | 3750 (77%) | 95.4 | 865 (43%) | 61.3 | 3483 (47%) | 78.4 | 6272 (80%) |
| *w.* `Sonata` (100 samples) | 63.3 | 17005 (100%) | 96.8 | 3883 (79%) | 95.0 | 1005 (50%) | 60.9 | 3994 (54%) | 79.0 | 6472 (83%) |

**Integration with Existing Methods.** A key advantage of `Sonata` is its compatibility with existing CoT optimization techniques. To validate this, we integrate `Sonata` with REFRAIN (Sun et al., 2025), a representative early-stopping method for CoT reasoning. Specifically, we first apply our `Sonata` adapter to decide whether to enable thinking; if thinking is enabled, REFRAIN is then applied during the generation phase to determine when to terminate reasoning early. Table 6 presents results on Qwen3-8B across all benchmarks. The combined approach achieves 78.7% average accuracy while reducing token usage to 64% of the vanilla model, representing an additional 15% token reduction compared to `Sonata` alone (79% token usage) with minimal accuracy drop. These results demonstrate that `Sonata` serves as an effective outer "when to think" controller that naturally composes with existing efficient reasoning methods, enabling further efficiency gains through orthogonal optimization.

Table 6: Integration of `Sonata` with REFRAIN early-stopping method. Results on Qwen3-8B across all benchmarks. Token percentages are relative to the vanilla model.

| Methods | AIME25 | | MATH-500 | | GSM8K | | GPQA | | Average | |
|---|---|---|---|---|---|---|---|---|---|---|
| | Acc. (↑) | #Tokens (↓) | Acc. (↑) | #Tokens (↓) | Acc. (↑) | #Tokens (↓) | Acc. (↑) | #Tokens (↓) | Acc. (↑) | #Tokens (↓) |
| Qwen3-8B | 60.0 | 16995 | 97.6 | 4900 | 95.2 | 1994 | 60.1 | 7458 | 78.2 | 7837 |
| *w.* Const. Budget | 30.0 | 4096 (24%) | 93.2 | 4096 (84%) | 95.0 | 4096 (205%) | 57.1 | 4096 (55%) | 68.8 | 4096 (52%) |
| *w.* Self-Judge | 60.0 | 17019 (100%) | 96.0 | 4315 (88%) | 92.7 | 1076 (54%) | 57.6 | 5913 (79%) | 76.6 | 7080 (90%) |
| *w.* Sonata | 63.3 | 16449 (97%) | 97.4 | 3694 (75%) | 95.6 | 890 (45%) | 62.0 | 3590 (48%) | 79.6 | 6156 (79%) |
| *w.* Sonata + REFRAIN | 60.0 | 12840 (76%) | 97.6 | 3309 (68%) | 95.0 | 845 (42%) | 61.8 | 2958 (40%) | 78.7 | 4988 (64%) |

**Multi-position and Multi-layer Aggregation.** To investigate whether aggregating information from multiple positions or layers could improve adapter performance, we evaluate two variants: (1) concatenating hidden states from the last token across the last 4 layers, and (2) concatenating hidden states from the last 4 tokens at the final layer. Table 7 presents results on Qwen3-8B across all benchmarks. The original `Sonata` design using only the last token from the last layer achieves the best overall performance with 79.6% average accuracy and 79% token usage. In contrast, the last-4-layers variant achieves 78.4% accuracy with 78% token usage, while the last-4-tokens variant shows significantly degraded performance at 71.1% accuracy. These results validate our design choice motivated by Figure 3, which demonstrates that self-consistency patterns are most distinguishable in the final layer. Incorporating information from multiple positions or layers appears to introduce noise rather than a beneficial signal.

Table 7: Ablation study on multi-position and multi-layer aggregation. Results on Qwen3-8B across all benchmarks. Token percentages are relative to the vanilla model.

| Methods | AIME25 | | MATH-500 | | GSM8K | | GPQA | | Average | |
|---|---|---|---|---|---|---|---|---|---|---|
| | Acc. (↑) | #Tokens (↓) | Acc. (↑) | #Tokens (↓) | Acc. (↑) | #Tokens (↓) | Acc. (↑) | #Tokens (↓) | Acc. (↑) | #Tokens (↓) |
| Qwen3-8B | 60.0 | 16995 | 97.6 | 4900 | 95.2 | 1994 | 60.1 | 7458 | 78.2 | 7837 |
| *w.* Const. Budget | 30.0 | 4096 (24%) | 93.2 | 4096 (84%) | 95.0 | 4096 (205%) | 57.1 | 4096 (55%) | 68.8 | 4096 (52%) |
| *w.* Self-Judge | 60.0 | 17019 (100%) | 96.0 | 4315 (88%) | 92.7 | 1076 (54%) | 57.6 | 5913 (79%) | 76.6 | 7080 (90%) |
| *w.* Sonata (last layer, last token) | 63.3 | 16449 (97%) | 97.4 | 3694 (75%) | 95.6 | 890 (45%) | 62.0 | 3590 (48%) | 79.6 | 6156 (79%) |
| *w.* Sonata (last 4 layers, last token) | 63.3 | 16449 (97%) | 95.8 | 3740 (76%) | 94.7 | 996 (50%) | 59.8 | 3302 (44%) | 78.4 | 6122 (78%) |
| *w.* Sonata (last layer, last 4 tokens) | 46.7 | 9549 (56%) | 88.2 | 2948 (60%) | 91.3 | 523 (26%) | 58.2 | 2840 (38%) | 71.1 | 3965 (51%) |

# 6 CONCLUSION

In this work, we investigate the adaptive allocation of thinking budgets in large language models (LLMs), revealing that self-consistency serves as a principled proxy for determining when and how much long chain-of-thought (CoT) reasoning is needed. By analyzing the relationship between self-consistency and thinking necessity, we show that queries with low self-consistency benefit significantly from extended reasoning, while high self-consistency queries require minimal or no thinking. This insight is further validated by our observation that self-consistency patterns are highly distinguishable within the latent space, enabling efficient prediction without expensive sampling. Leveraging these findings, we propose `Sonata`, a lightweight, offline-trained adapter that predicts self-consistency directly from query hidden representations during the prefilling stage. `Sonata` dynamically allocates thinking budgets on-the-fly, introducing negligible computational overhead ($< 1‰$) while being generalizable across diverse tasks without task-specific fine-tuning. Experimental results across multiple models (Qwen3-8B, Qwen3-32B, GPT-OSS-120B, Qwen3-235B-A22B) and challenging benchmarks (AIME25, GSM8K, MATH500, GPQA, Live-CodeBench) demonstrate that Sonata achieves up to 60% reduction in thinking tokens while maintaining or improving accuracy by up to 2%. Importantly, `Sonata` is compatible with existing CoT compression techniques, enabling further efficiency gains when combined with these methods. Our approach offers a practical and interpretable solution for optimizing test-time compute in reasoning models, paving the way for more efficient deployment of thinking LLMs at scale.

## ETHICS STATEMENT

This work focuses on improving the computational efficiency of large language models during inference, which we believe contributes positively to making AI systems more accessible and environmentally sustainable by reducing computational resource requirements. Our research does not involve human subjects, and all experiments use publicly available benchmarks (AIME25, MATH-500, GSM8K, GPQA, LiveCodeBench) that contain no sensitive personal information. The adaptive thinking mechanism we propose does not introduce discriminatory biases and treats all queries based solely on their intrinsic reasoning complexity. We have no conflicts of interest to declare, and this work was conducted without external funding that could influence our findings. We believe our research adheres to the ICLR Code of Ethics and poses no foreseeable risks to individuals or society.

## REPRODUCIBILITY STATEMENT

To ensure reproducibility of our results, we provide comprehensive implementation details throughout the paper and supplementary materials. The core algorithms for our adapter training and inference are detailed in Algorithms 1 and 2. All experiments use publicly available models from the Qwen3 family and GPT-OSS, with model identifiers and access instructions provided in Section 5. Our calibration dataset construction process, using 1000 problems from the OpenMathReasoning dataset, is described in Section 5. Exact hyperparameters including sampling parameters, number of samples for self-consistency computation, and threshold settings are specified throughout Section 5. The adapter architecture (two-layer MLP) and training procedure are detailed in Section 4. All experiments were conducted on NVIDIA B200 GPUs with reproducibility ensured through fixed random seeds $\{233, 234, 235\}$. Evaluation metrics and benchmark details are provided in Section 5, with results averaged over three trials for statistical reliability.

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

# APPENDIX

## A  THE USE OF LARGE LANGUAGE MODELS (LLMS)

To improve readability, we utilized Anthropic's Claude Opus 4.1 exclusively as a language polishing tool. We use it for grammar correction, proofreading, and stylistic refinement. It did not contribute to the generation of any scientific content or ideas, and its usage is consistent with standard practices for scientific writing.

## B  TECHNICAL DETAILS

### B.1  ADAPTER ARCHITECTURE DETAILS

Our adapter employs a 2-layer MLP architecture designed to map high-dimensional hidden representations to *self-consistency* predictions. This architecture uses only $64$ hidden units to minimize inference overhead.

**Pseudocode.**   We show some pseudocode to demonstrate the implementation of our proposed `Sonata` adapter in JAX style.

```
def mlp_predictor(params, x):
    """Two-layer MLP for self-consistency prediction.

    Args:
        params: {'w1': (d, 64), 'b1': (64,),
                 'w2': (64, 1), 'b2': (1,)}
        x: Hidden states of shape (d,)
    Returns:
        Self-consistency prediction in [0, 1]
    """
    # First layer with GELU
    h = gelu(x @ params['w1'] + params['b1'])
    h = dropout(h, rate=0.1) if training else h

    # Output layer with sigmoid
    y = sigmoid(h @ params['w2'] + params['b2'])

    return y
```

**Training Details.**   The adapter is trained using MSE loss between predicted and ground-truth self-consistency scores. We employ Xavier uniform initialization for weights and zero initialization for biases to ensure stable training. Training uses AdamW optimizer with learning rate $10^{-5}$ and weight decay $10^{-5}$. We train with batch size 16 and employ a linear learning rate scheduler that decays from $10^{-5}$ to $10^{-6}$ over the training period. Gradient clipping is applied to ensure stable training. The total number of its parameters is $(d \times 64) + 64 + (64 \times 1) + 1$, which for Qwen3-8B ($d = 4096$) consumes around 262K parameters, negligible compared to the 8B parameters of the base LLM. This results in less than $0.1\%$ additional FLOPs compared to a single transformer layer forward pass, validating our claim of negligible computational overhead.

### B.2  SELF-JUDGE

The *self-judge* baseline asks LLM to assess its own need for extended CoT reasoning before generating a response. This method involves prompting the model to make a binary decision about whether to engage its thinking capability.

Specifically, for each query `QUESTION`, we first prompt the model with:

```
Analyze the following question and determine if it requires
very long step-by-step thinking for you to solve correctly:

Question:  {QUESTION}

Does this question require very long, complex thinking?
Answer with only 'YES' or 'NO'.
```

Based on the model's response, we proceed as follows: ❶ If the model responds "YES", we enable thinking mode by allowing the model to generate chain-of-thought tokens between `<think>` and `</think>` tags before producing the final answer. ❷ If the model responds "NO", we enforce immediate termination of thinking by inserting `</think>` directly after `<think>`, forcing the model to generate the answer without intermediate reasoning steps.

Notably, this baseline requires an additional forward pass for the self-assessment, incurring approximately 100-200 tokens of prefilling overhead per query for the judgment prompt and response.

## C  ADDITIONAL EXPERIMENTAL RESULTS

### C.1  EXTENDED SONATA WITH THINKING GAIN PREDICTION

We evaluate an extended version of `Sonata` that predicts both self-consistency and thinking gain together. We train two separate adapters: one for self-consistency prediction (as in the original `Sonata`) and another for thinking gain prediction. Specifically, the extended adapters only enable thinking if predicted self-consistency $< 0.3$ *and* predicted thinking gain $> 0.1$; otherwise, proceed without thinking.

Table 8 presents results on Qwen3-8B across all benchmarks. The extended `Sonata` achieves $79.3\%$ average accuracy with $80\%$ token usage, nearly identical to the original `Sonata` ($79.6\%$ accuracy, $79\%$ token usage). This negligible difference validates our observation that intrinsically difficult queries are rare in practice and have minimal impact on overall performance. The original binary self-consistency-based approach is therefore sufficient for practical deployment.

Table 8: Comparison of original `Sonata` with extended version that predicts both self-consistency and thinking gain. Results on Qwen3-8B across all benchmarks.

| Methods | AIME25 | | MATH-500 | | GSM8K | | GPQA | | Average | |
|---|---|---|---|---|---|---|---|---|---|---|
| | Acc. (↑) | #Tokens (↓) | Acc. (↑) | #Tokens (↓) | Acc. (↑) | #Tokens (↓) | Acc. (↑) | #Tokens (↓) | Acc. (↑) | #Tokens (↓) |
| Qwen3-8B | 60.0 | 16995 | 97.6 | 4900 | 95.2 | 1994 | 60.1 | 7458 | 78.2 | 7837 |
| *w.* Sonata (self-consistency) | 63.3 | 16449 (97%) | 97.4 | 3694 (75%) | 95.6 | 890 (45%) | 62.0 | 3590 (48%) | 79.6 | 6156 (79%) |
| *w.* Sonata (self-consistency + thinking gain) | 63.3 | 16985 (100%) | 97.2 | 3650 (74%) | 95.0 | 905 (45%) | 61.7 | 3483 (47%) | 79.3 | 6256 (80%) |

### C.2  FINE-GRAINED SELF-JUDGE BASELINE

To ensure a fair comparison with self-judge baselines, we evaluate a fine-grained 5-level difficulty rating prompt in addition to the binary version presented in the main paper. Specifically, the model first predicts a difficulty score $S \in \{1, 2, 3, 4, 5\}$, then allocates a thinking budget of $(S-1) \times 2048$ tokens, where $S = 1$ corresponds to no thinking.

Table 9 presents results on Qwen3-8B across all benchmarks. The 5-level self-judge achieves $76.3\%$ average accuracy with $94\%$ token usage, underperforming both the binary self-judge ($76.6\%$ accuracy, $90\%$ token usage) and `Sonata` ($79.6\%$ accuracy, $79\%$ token usage). The degraded performance suggests that fine-grained budget allocation without training or adaptation can be challenging for models to execute reliably.

### C.3  FINE-GRAINED THINKING CONTROL

As an early attempt to explore whether more granular thinking budget allocation could improve performance with our method, we extend `Sonata` from binary control to 4-level control with thresh-

Table 9: Comparison of binary and 5-level fine-grained self-judge baselines. Results on Qwen3-8B across all benchmarks.

| Methods | AIME25 | | MATH-500 | | GSM8K | | GPQA | | Average | |
|---|---|---|---|---|---|---|---|---|---|---|
| | Acc. (↑) | #Tokens (↓) | Acc. (↑) | #Tokens (↓) | Acc. (↑) | #Tokens (↓) | Acc. (↑) | #Tokens (↓) | Acc. (↑) | #Tokens (↓) |
| Qwen3-8B | 60.0 | 16995 | 97.6 | 4900 | 95.2 | 1994 | 60.1 | 7458 | 78.2 | 7837 |
| *w.* Self-Judge (binary) | 60.0 | 17019 (100%) | 96.0 | 4315 (88%) | 92.7 | 1076 (54%) | 57.6 | 5913 (79%) | 76.6 | 7080 (90%) |
| *w.* Self-Judge (5-level) | 60.0 | 16990 (100%) | 94.6 | 4323 (88%) | 94.3 | 2350 (118%) | 56.1 | 5857 (79%) | 76.3 | 7380 (94%) |
| *w.* Sonata | 63.3 | **16449 (97%)** | 97.4 | **3694 (75%)** | 95.6 | **890 (45%)** | 62.0 | **3590 (48%)** | 79.6 | **6156 (79%)** |

olds $\{0.3, 0.5, 0.7, 0.9\}$ corresponding to thinking modes: {non-thinking (0 tokens), low thinking ($< 1024$ tokens), medium thinking ($< 4096$ tokens), high thinking (unlimited tokens)}.

Table 10 presents results on Qwen3-8B across all benchmarks. The 4-level Sonata achieves 78.4% average accuracy with 75% token usage, slightly underperforming the binary version (79.6% accuracy, 79% token usage). The degraded performance suggests that fine-grained control introduces brittleness due to complex threshold combinations that are difficult to tune manually. Future work could explore data-driven or training-based methods to automatically optimize thresholds for fine-grained thinking control, though our results validate the simplicity and effectiveness of the binary design for practical deployment.

Table 10: Comparison of binary and 4-level fine-grained thinking control. Results on Qwen3-8B across all benchmarks.

| Methods | AIME25 | | MATH-500 | | GSM8K | | GPQA | | Average | |
|---|---|---|---|---|---|---|---|---|---|---|
| | Acc. (↑) | #Tokens (↓) | Acc. (↑) | #Tokens (↓) | Acc. (↑) | #Tokens (↓) | Acc. (↑) | #Tokens (↓) | Acc. (↑) | #Tokens (↓) |
| Qwen3-8B | 60.0 | 16995 | 97.6 | 4900 | 95.2 | 1994 | 60.1 | 7458 | 78.2 | 7837 |
| *w.* Const. Budget | 30.0 | 4096 (24%) | 93.2 | 4096 (84%) | 95.0 | 4096 (205%) | 57.1 | 4096 (55%) | 68.8 | 4096 (52%) |
| *w.* Self-Judge | 60.0 | 17019 (100%) | 96.0 | 4315 (88%) | 92.7 | 1076 (54%) | 57.6 | 5913 (79%) | 76.6 | 7080 (90%) |
| *w.* Sonata (binary) | 63.3 | **16449 (97%)** | 97.4 | **3694 (75%)** | 95.6 | **890 (45%)** | 62.0 | **3590 (48%)** | 79.6 | **6156 (79%)** |
| *w.* Sonata (4-level) | 63.3 | 16449 (97%) | 96.2 | 3258 (66%) | 95.4 | 853 (43%) | 58.8 | 2983 (40%) | 78.4 | 5886 (75%) |

## C.4 Examples around Decision Threshold $\tau_0 = 0.3$

We empirically determined $\tau_0 = 0.3$ as our decision threshold by evaluating values in $\{0.1, 0.3, 0.5\}$. We found that $\tau_0 = 0.1$ resulted in significant performance degradation, while $\tau_0 = 0.5$ led to excessive token consumption. The threshold $\tau_0 = 0.3$ represents an optimal balance between accuracy and efficiency. To further illustrate the effectiveness of this threshold, we provide two examples from MATH-500 on Qwen3-8B with their predicted self-consistency scores:

**Example 1** (adapter score = 0.25, below threshold): "Let $p(x)$ be a polynomial of degree 5 such that $p(n) = \frac{n}{n^2-1}$ for $n = 2, 3, 4, \ldots, 7$. Find $p(8)$."

**Example 2** (adapter score = 0.42, above threshold): "The proper divisors of 12 are 1, 2, 3, 4 and 6. A proper divisor of an integer $N$ is a positive divisor of $N$ that is less than $N$. What is the sum of the proper divisors of the sum of the proper divisors of 284?"

Both examples are labeled as difficulty level 5 (hardest) in MATH-500. However, Example 1 requires abstract algebraic reasoning and polynomial-construction techniques typically requiring competition mathematics experience, while Example 2 primarily involves arithmetic and divisor fluency with more straightforward reasoning.

Empirically, over 3 trials per example, Qwen3-8B achieves pass@3 = 0 without thinking and pass@3 = 1 with thinking on Example 1, while achieving pass@3 = 1 for both modes on Example 2. Our Sonata policy with $\tau_0 = 0.3$ correctly answers both examples while avoiding unnecessary thinking cost on Example 2 and preserving necessary thinking for Example 1.

## C.5 Model Outputs Before and After Applying Sonata

To clarify potential misunderstandings, Sonata does not change the model's reasoning dynamics or content. It only shifts between thinking mode (with intermediate reasoning) and non-thinking mode (without intermediate reasoning). We further provide a concrete example from MATH-500:

**Input Query:** "How many positive whole-number divisors does 196 have?"

**Output Before Applying `Sonata`** (with thinking enabled, abbreviated for space):

```
<think>
Okay, so I need to figure out how many positive whole-number divisors 196 has. Hmm,
divisors... right. Let me recall. Divisors are numbers that divide another number without
leaving a remainder. [... extensive reasoning process omitted ...] Therefore, the final answer
is: \boxed{9}. I hope it is correct.
</think>

To determine how many positive whole-number divisors the number 196 has, we can
use the **prime factorization method** combined with a well-known divisor-counting
formula. [... detailed explanation ...] Therefore, the final answer is: \boxed{9}. I hope it is
correct.
```

**Output After Applying `Sonata`** (thinking disabled, direct answer):

```
<think>

</think>

To find how many **positive whole-number divisors** the number **196** has, we
follow these steps: [... direct solution without intermediate reasoning ...] Therefore, the
number of positive whole-number divisors of 196 is: \boxed{9}. Therefore, the final answer
is: \boxed{9}. I hope it is correct.
```

Both outputs produce the correct answer, but `Sonata` eliminates the unnecessary thinking tokens for this straightforward arithmetic problem, resulting in significant computing savings.

## D  DISCUSSION

**Future Directions.**   While our current work focuses on deciding *whether to think* before any reasoning tokens are generated, an interesting extension would be to dynamically adjust reasoning length *during* ongoing long-form reasoning based on self-consistency signals. This would require (1) a richer controller that can operate across larger hidden spaces beyond our current single-hidden-state MLP, and (2) a potentially more complex training setup, either through substantial additional offline data with early-stopped reasoning traces to supervise fine-grained stopping policies, or through online reinforcement learning that directly optimizes early-stopping controllers under token budget-performance trade-offs. Both directions represent promising avenues for future research in adaptive test-time compute allocation.

