# OpenReview forum: "Adaptive Thinking: Large Language Models Know When to Think in Latent Space"
_ICLR.cc/2026/Conference — ICLR 2026 Poster_

### Official Review · Reviewer_sPcD · 2025-10-24

**Soundness:** 4
**Presentation:** 3
**Contribution:** 3
**Rating:** 4
**Confidence:** 4

**Summary:**

This paper uses self-consistency—the agreement among multiple reasoning paths for the same problem—as an operational proxy for determining “whether long reasoning is needed.” Empirically, samples with lower self-consistency tend to require extended reasoning. Building on this observation, the paper proposes a highly lightweight gating scheme, Sonata: during the prefill stage, it predicts self-consistency directly from the hidden representation of the last layer and the last position, and accordingly makes an instant decision on whether and how much to think before decoding. The adapter is trained offline on a calibration set, has an online cost of less than one-thousandth, can be reused across tasks, and can be combined with existing CoT compression methods for further gains. Experiments across multiple models and benchmarks show that it can reduce reasoning tokens by 20%–60% without loss of accuracy, or improve accuracy by about 2% under the same token budget, thus advancing the Pareto frontier of “performance–efficiency.” Overall, the paper turns the decision of “whether and how much to think” from a heuristic into a learnable, deployable, and almost zero-overhead paradigm, balancing theoretical motivation and engineering practicality.

**Strengths:**

1. The problem addressed in this paper is crucial for the real-world deployment of large models. The Pareto frontier results clearly demonstrate that the approach achieves higher accuracy under the same compute budget or lower cost under the same accuracy, providing intuitive and persuasive evidence.

2. The paper innovatively observes that “self-consistency becomes more separable in deeper hidden spaces,” and accordingly predicts self-consistency using the hidden state of the last layer and last token to form a lightweight gating mechanism. The idea is novel and well supported by extensive experiments.

3. The method is simple and practical: the adapter is trained once offline via calibration using a two-layer MLP, and during inference only reads one hidden state in the prefill stage for gating—introducing almost zero additional overhead and being highly engineering-friendly.

**Weaknesses:**

1. The coverage of baselines could be further expanded. Although comparisons with two entropy-based baselines are sufficient, the study does not include some of the stronger recent uncertainty or calibration signals (e.g., multi-step verification-based early stopping, lightweight proxies based on draft diversity, or learned “thinking switch” controllers). The current conclusion mainly shows that “entropy signals are inadequate,” without directly competing against the strongest adaptive reasoning strategies.

2. In Figure 4, the authors mention that when $s<\tau$, the model enters the reasoning phase and adaptively allocates reasoning token budget. However, the main text does not explain how these different tiers are assigned, and the algorithmic framework only shows a binary classification. The authors should report the concrete procedure for adaptive budget allocation in detail.

**Questions:**

1. The paper only discusses using the hidden state of the “last layer and last token.” Would multi-position aggregation or multi-layer fusion bring further benefits? This limitation might affect robustness on semantically complex or multi-turn inputs.

2. The current method decides the reasoning length solely based on the state of the last token of the question. Could this be extended to dynamically determine, during long-form reasoning, whether the reasoning token budget can be shortened according to the proposed self-consistency measure?

---

> ### Author Response · Authors · 2025-11-22
> **Response to Reviewer sPcD (1/3)**
>
> Dear Reviewer sPcD,
>
> We thank Reviewer sPcD for recognizing the importance ("crucial for the real-world deployment"), innovation ("intuitive and persuasive evidence", "novel and well supported by extensive experiments"), and practicability ("highly engineering-friendly", "simple and practical") behind our work on adaptive thinking. To answer Reviewer sPcD's questions, we provide point-to-point responses below.

---

> ### Author Response · Authors · 2025-11-22
> **Response to Reviewer sPcD (2/3)**
>
> ### [W1. Baseline coverage and relation to recent adaptive reasoning methods.]
>
> We would like to clarify that our goal is not to replace more recent adaptive reasoning techniques, but to provide a complementary "when to think" controller that can be composed with them. Our method is "compatible" with all of these recent uncertainty or calibration signal-based early-stopping methods, as illustrated in Section 1 (around line 70), and mentioned by Reviewer zM6i/SHJi/EUV3.
>
> To address your concern more concretely, we have added experiments that combine Sonata with REFRAIN **[1]**, a representative early-stopped thinking method. Concretely, we first use Sonata to decide whether to think or not; if Sonata decides to think, REFRAIN is then applied during the thinking phase to determine when to stop. As shown in **Table R1** (and included in the revised draft), this composition reduces the number of generated tokens by about 15% on average while maintaining essentially the same accuracy. This demonstrates that Sonata can be straightforwardly integrated with recent adaptive reasoning methods (dynamic budget / early stopping / CoT-pruning), and supports our claim that Sonata serves as an effective outer "when to think" controller that complements, rather than competes with, these methods.
>
> **[Table R1] (Accuracy; # of tokens)**
>
> |                                  | AIME25             | MATH-500         | GSM8K             | GPQA             | Average          |
> | -------------------------------- | ------------------ | ---------------- | ----------------- | ---------------- | ---------------- |
> | Qwen3-8B                         | 60.0; 16995        | 97.6; 4900       | 95.2; 1994        | 60.1; 7458       | 78.2; 7837       |
> | w. Const Budget                  | 30.0; 4096 (24%)   | 93.2; 4096 (84%) | 95.0; 4096 (205%) | 57.1; 4096 (55%) | 68.8; 4096 (52%) |
> | w. Self-Judge (binary, original) | 60.0; 17019 (100%) | 96.0; 4315 (88%) | 92.7; 1076 (54%)  | 57.6; 5913 (79%) | 76.6; 7080 (90%) |
> | w. Sonata                        | 63.3; 16449 (97%)  | 97.4; 3694 (75%) | 95.6; 890 (45%)   | 62.0; 3590 (48%) | 79.6; 6156 (79%) |
> | w. Sonata + REFRAIN              | 60.0; 12840 (76%)  | 97.6; 3309 (68%) | 95.0; 845 (42%)   | 61.8; 2958 (40%) | 78.7; 4988 (64%) |
>
> ### [W2. Concrete procedure for adaptive budget allocation in detail.]
>
> To clarify, our "concrete procedure for adaptive budget allocation" is already presented in Algorithm 2 in our original draft. The current adaptive reasoning token budgets to allocate are either 0 or unlimited (i.e. no constraint), exploring different tiers upon Sonata adapter could be one interesting future research work.
>
> ### [Q1. Multi-position aggregation or multi-layer fusion.]
>
> The motivation for why we use the last token and the last layer's hidden states is presented in Figure 3 and Section 3.2 in our original draft, as these hidden states are more distinguishable in terms of self-consistency. To answer your question, we conduct an additional ablation study of Sonata with 1) concatenated hidden states of the last token from the last 4 layers, and 2) concatenated hidden states of the last 4 tokens from the last layer, on Qwen3-8B model across all tasks. As shown by the results in **Table R2**, the original Sonata with the last token hidden states from the last layers consistently outperforms the other two variants 1) and 2) in terms of both accuracy and efficiency, validating our conclusion. This is potentially because incorporating multi-position or multi-layer hidden states could introduce more noise, instead of helping self-consistency to be more distinguishable. We further include these results and discussion in our revised draft.
>
> **[Table R2] (Accuracy; # of tokens)**
>
> |                                              | AIME25             | MATH-500         | GSM8K             | GPQA             | Average           |
> | -------------------------------------------- | ------------------ | ---------------- | ----------------- | ---------------- | ----------------- |
> | Qwen3-8B                                     | 60.0; 16995        | 97.6; 4900       | 95.2; 1994        | 60.1; 7458       | 78.2; 7837        |
> | w. Const Budget                              | 30.0; 4096 (24%)   | 93.2; 4096 (84%) | 95.0; 4096 (205%) | 57.1; 4096 (55%) | 68.8; 4096 (52%)  |
> | w. Self-Judge                                | 60.0; 17019 (100%) | 96.0; 4315 (88%) | 92.7; 1076 (54%)  | 57.6; 5913 (79%) | 76.6; 7080 (90%)  |
> | w. Sonata (last layer, last token, original) | 63.3; 16449 (97%)  | 97.4; 3694 (75%) | 95.6; 890 (45%)   | 62.0; 3590 (48%) | 79.6; 6156 (79%)  |
> | w. Sonata (last 4 layers, last token)        | 63.3; 16449 (97%)  | 95.8; 3740 (76%) | 94.7; 996 (50%)   | 59.8; 3302 (44%) | 78.4; 6122 (78%)  |
> | w. Sonata (last layer, last 4 tokens)        | 46.7; 9549 (56%)   | 88.2; 2948 (60%) | 91.3; 523 (26%)   | 58.2; 2840 (38%) | 71.1%; 3965 (51%) |

---

> ### Author Response · Authors · 2025-11-22
> **Response to Reviewer sPcD (3/3)**
>
> ### [Q2. Dynamic reasoning length decision along long-form reasoning.]
>
> Thanks for the question. In our current design, as described in Section 4.1 of the original draft, Sonata uses the hidden state at a specific position and layer to make the decision: we take the hidden state of the last token of the query (which is always a newline "\n") from a fixed transformer layer, and feed it into a lightweight 2-layer MLP to predict the thinking score. In other words, the current adapter is designed only to decide "whether to think at all" before any reasoning tokens are generated, rather than dynamically adjusting the length of an ongoing reasoning trace.
>
> Extending the idea to dynamically shorten long-form reasoning based on a self-consistency signal during thinking is an interesting research question. Doing so would likely require 1) a richer controller across larger hidden space than our current small MLP on a single hidden state, and 2) a more complex training setup. For example, one would need either 1) substantial additional offline data with early-stopped reasoning traces to supervise a fine-grained stopping policy, or 2) an online RL process that directly optimizes an early-stopping controller under a token budget–performance trade-off. Both directions go beyond the scope of our current work, which focuses on a simple and practical "when to think" gate that can be easily plugged into existing models.
>
> We add a discussion of this idea and potential extensions into the revised draft, and we plan to explore it as an important direction for future work.

---

> > ### Comment · Reviewer_sPcD · 2025-11-24
> >
> > Thank you for your response. I believe you have addressed most of my concerns, and I have accordingly raised my score.

---

> ### Author Response · Authors · 2025-11-24
>
> Dear Reviewer sPcD,
>
> Thank you so much for all the constructive feedback and positive evaluations. We are again very thankful for your time and help, and for raising your score!

---

### Official Review · Reviewer_EUV3 · 2025-10-30

**Soundness:** 2
**Presentation:** 3
**Contribution:** 2
**Rating:** 4
**Confidence:** 4

**Summary:**

The paper proposes Sonata, a lightweight plug-in designed to predict answer self-consistency from the last-layer hidden state of a large language model (LLM). The predicted score determines whether the model should enable long chain-of-thought (CoT) reasoning. Trained offline on 1,000 competition-math problems, the approach introduces less than 0.1% additional FLOPs while reducing CoT token usage by 20–60%, maintaining or slightly improving accuracy on four math and science benchmarks.

**Strengths:**

1. The proposed method is highly modular and can be seamlessly integrated into existing large language models without modifying or fine-tuning the base parameters. It is also compatible with current chain-of-thought (CoT) compression and optimization techniques, enabling straightforward adoption in practical systems. Moreover, the approach incurs negligible computational cost, requiring only a single MLP forward pass on already-computed hidden states.
2. The authors provide empirical evidence that latency and memory overhead remain minimal across a wide range of model scales (from 8B to 235B parameters).
3. The paper is overall well-written and easy to follow.

**Weaknesses:**

1. The preliminary visualization experiment, which performs PCA clustering on the hidden state of the final token, is conducted only on a single mathematical benchmark. The study does not extend this analysis to other datasets or task types, such as commonsense reasoning or code generation. It remains unclear whether the observed separability is a general phenomenon or specific to mathematical domains.

2. The authors claim that the MLP module trained to predict self-consistency on math datasets can generalize across domains. However, among the four benchmarks used for evaluation, three are mathematical (AIME, MATH, GSM) and only one is scientific (GPQA). This experimental setup provides limited evidence for cross-domain generalization.

3. The paper fixes the decision threshold for enabling reasoning at τ = 0.3, but does not justify how this value is chosen. Providing illustrative cases that show differences in question difficulty or prediction confidence around this threshold would make the calibration choice more convincing.

4. The paper lacks a qualitative case study demonstrating the effect of Sonata on model behavior. For instance, it would be helpful to show concrete examples comparing model outputs before and after applying Sonata, to better understand how the method changes reasoning dynamics.

**Questions:**

No further questions beyond those discussed above.

---

> ### Author Response · Authors · 2025-11-22
> **Response to Reviewer EUV3 (1/3)**
>
> Dear Reviewer EUV3,
>
> Thanks to Reviewer EUV3 for recognizing the technical advantage of our proposed method for being "highly modular", "seamlessly integrated into existing LLMs", with "negligible computational cost", and for the writing of our paper for being "well-written and easy to follow". To address Reviewer EUV3's questions, we provide point-to-point response below.

---

> ### Author Response · Authors · 2025-11-22
> **Response to Reviewer EUV3 (2/3)**
>
> ### [W1. Preliminary visualization experiments.]
>
> Our intuition and proposed method are "generalizable" across domains, as illustrated in the Section 5.1 in our draft and mentioned by Reviewer zM6i. We further extend the PCA visualization analysis of Figure 3 in our draft on GPQA, a commonsense reasoning task, and include it in our revised draft. As shown in the **Figure 3** in our revised draft, PCA visualization on both math (existing) and commonsense reasoning (new) tasks validate our observation that self-consistency is highly distinguishable in deeper layers, particularly in deeper layers, as stated in Section 3.2 in our original draft.
>
> ### [W2. Cross-domain generalization.]
>
> Sonata is generalizable and task-agnostic, as illustrated in the results of GPQA in Section 5.2. We further conduct comparison experiments across baselines on the LiveCodeBench task (code generation) with Qwen3-8B and Qwen3-32B models. As shown by the LiveCodeBench results in **Table R1** and **R2**, Sonata consistently outperforms all other baselines on all tasks, with the highest pass@1 across all methods and ~10% reduced numbers of tokens compared to the vanilla models. This further validates the cross-domain generalization of our proposed Sonata. We further include the new task in our revised draft.
>
> **[Table R1] (Accuracy; # of tokens)**
>
> |                 | AIME25             | MATH-500         | GSM8K             | GPQA             | LiveCodeBench      | Average           |
> | --------------- | ------------------ | ---------------- | ----------------- | ---------------- | ------------------ | ----------------- |
> | Qwen3-8B        | 60.0; 16995        | 97.6; 4900       | 95.2; 1994        | 60.1; 7458       | 57.8; 14421        | 74.1; 9154        |
> | w. Const Budget | 30.0; 4096 (24%)   | 93.2; 4096 (84%) | 95.0; 4096 (205%) | 57.1; 4096 (55%) | 49.3; 4096 (28%)   | 64.9; 4096 (45%)  |
> | w. Self-Judge   | 60.0; 17019 (100%) | 96.0; 4315 (88%) | 92.7; 1076 (54%)  | 57.6; 5913 (79%) | 57.1%; 13496 (93%) | 72.7%; 8364 (91%) |
> | w. Sonata       | 63.3; 16449 (97%)  | 97.4; 3694 (75%) | 95.6; 890 (45%)   | 62.0; 3590 (48%) | 58.2; 13054 (90%)  | 75.3; 7535 (82%)  |
>
> **[Table R2] (Accuracy; # of tokens)**
>
> |                 | AIME25             | MATH-500          | GSM8K             | GPQA             | LiveCodeBench     | Average          |
> | --------------- | ------------------ | ----------------- | ----------------- | ---------------- | ----------------- | ---------------- |
> | Qwen3-32B       | 70.0; 14971        | 97.6; 3670        | 94.8; 1654        | 63.6; 5329       | 63.8; 13729       | 78.0; 7853       |
> | w. Const Budget | 56.7; 4096 (27%)   | 95.8; 4096 (112%) | 95.3; 4096 (248%) | 64.1; 4096 (77%) | 47.0; 4096 (30%)  | 69.8; 4096 (52%) |
> | w. Self-Judge   | 73.3; 15625 (104%) | 96.2; 3107 (85%)  | 93.1; 666 (40%)   | 54.5; 3807 (71%) | 59.3; 13360 (97%) | 75.3; 7313 (93%) |
> | w. Sonata       | 70.0; 14890 (100%) | 98.0; 2583 (70%)  | 94.4; 728 (44%)   | 63.1; 3568 (70%) | 63.4; 13600 (99%) | 77.8; 7074 (90%) |
>
> ### [W3. Decision threshold choice.]
>
> As described at the footnote in page 6 of our draft, empirically, "we find that τ0=0.3 generally works well". Specifically, we tried {0.1, 0.3, 0.5} and found that τ0=0.1 suffers from a significant performance drop and τ0=0.5 tends to consume much more tokens, while τ0=0.3 is a sweet spot that both maintains performance and reduces tokens effectively.
>
> To answer your question for "illustrative cases that show differences in question difficulty or prediction confidence around this threshold," we provide two examples from the MATH-500 benchmark on Qwen3-8B, which are also included in our revised draft along with corresponding discussions:
>
> -   **Example 1 (adapter score = 0.25):** "Let $p(x)$ be a polynomial of degree 5 such that\n\[p(n) = \frac{n}{n^2 - 1}\]for $n = 2,$ 3, 4, $\dots,$ 7. Find $p(8).$"
> -   **Example 2 (adapter score = 0.42):** "The proper divisors of 12 are 1, 2, 3, 4 and 6. A proper divisor of an integer $N$ is a positive divisor of $N$ that is less than $N$. What is the sum of the proper divisors of the sum of the proper divisors of 284?"
>
> Both examples are labeled as difficulty level 5 (hardest) in MATH-500. However, Example 1 tests abstract algebraic reasoning and polynomial-construction techniques, which typically require some math competition experience, whereas Example 2 is mainly about arithmetic and fluency with factors/divisors, and its reasoning is much more straightforward.
>
> Empirically, over 3 repeated trials per example, Qwen3-8B achieves pass@3 = 0 without thinking and pass@3 = 1 with thinking on Example 1, while it achieves pass@3 = 1 for both thinking and non-thinking on Example 2. In this setting, our Sonata policy with tau_0 = 0.3 correctly answers both examples, while avoiding unnecessary thinking cost on Example 2 and preserving the necessary thinking content for Example 1.

---

> ### Author Response · Authors · 2025-11-22
> **Response to Reviewer EUV3 (3/3)**
>
> ### [W4. Examples comparing model outputs before and after applying Sonata.]
>
> Our proposed Sonata does NOT change the model's reasoning dynamics/content themselves, as it only shifts between thinking (with intermediate reasoning) and non-thinking (without intermediate reasoning) modes. This is also illustrated in Algorithm 2 in our original draft. To answer your question about "concrete examples comparing model outputs before and after applying Sonata", we provide one concrete example here and more of them in our revised draft, selected from MATH-500 dataset.
>
> -   **Input:** "How many positive whole-number divisors does 196 have?"
> -   **Output before applying Sonata** (a large mediate portion of thinking content between `<think>` and `</think>` is omitted and marked with "..." due to response length constraint): "`<think>\nOkay, so I need to figure out how many positive whole-number divisors 196 has. Hmm, divisors... right. Let me recall. Divisors are numbers that divide another number without leaving a remainder. So, for example, the divisors of 6 are 1, 2, 3, 6. But how do I find them all for 196 efficiently?\n\nI think the standard method is to start by prime factorizing the number. Once I have the prime factors, there's a formula to calculate the number of divisors. Let me remember…… Therefore, the final answer is: $\\boxed{9}$. I hope it is correct.\n</think>\n\nTo determine how many positive whole-number divisors the number 196 has, we can use the **prime factorization method** combined with a well-known divisor-counting formula.\n\n---\n\n### Step 1: Prime Factorization of 196\n\nWe begin by breaking down 196 into its prime factors:\n\n- $ 196 \\div 2 = 98 $\n- $ 98 \\div 2 = 49 $\n- $ 49 \\div 7 = 7 $\n- $ 7 \\div 7 = 1 $\n\nSo, the prime factorization of 196 is:\n\n$$\n196 = 2^2 \\times 7^2\n$$\n\n---\n\n### Step 2: Apply the Divisor Formula\n\nIf a number $ n $ has the prime factorization:\n\n$$\nn = p_1^{a_1} \\times p_2^{a_2} \\times \\dots \\times p_k^{a_k}\n$$\n\nThen the total number of **positive divisors** of $ n $ is given by:\n\n$$\n(a_1 + 1)(a_2 + 1)\\dots(a_k + 1)\n$$\n\nIn our case:\n\n- $ a_1 = 2 $ (for the prime factor 2)\n- $ a_2 = 2 $ (for the prime factor 7)\n\nSo the number of positive divisors is:\n\n$$\n(2 + 1)(2 + 1) = 3 \\times 3 = 9\n$$\n\n---\n\n### Step 3: Verification by Listing Divisors\n\nTo double-check, we can list all the positive divisors of 196:\n\n1, 2, 4, 7, 14, 28, 49, 98, 196\n\nCounting these gives exactly **9** divisors, confirming our result.\n\n---\n\n### Final Answer\n\nTherefore, the final answer is: $\\boxed{9}$. I hope it is correct.`"
> -   **Output after applying Sonata:** "`<think>\n\n</think>\n\nTo find how many **positive whole-number divisors** the number **196** has, we follow these steps:\n\n---\n\n### **Step 1: Prime Factorization of 196**\n\nWe begin by factoring 196 into its **prime factors**.\n\n- 196 is even, so divide by 2: \n $ 196 \\div 2 = 98 $ \n- 98 is also even: \n $ 98 \\div 2 = 49 $ \n- 49 is not divisible by 2, but it is divisible by 7: \n $ 49 \\div 7 = 7 $ \n- And 7 is a prime number.\n\nSo, the prime factorization of 196 is:\n\n$$\n196 = 2^2 \\times 7^2\n$$\n\n---\n\n### **Step 2: Use the Divisor Formula**\n\nIf a number $ n $ has a prime factorization of the form:\n\n$$\nn = p_1^{e_1} \\times p_2^{e_2} \\times \\dots \\times p_k^{e_k}\n$$\n\nThen the total number of **positive divisors** of $ n $ is:\n\n$$\n(e_1 + 1)(e_2 + 1)\\dots(e_k + 1)\n$$\n\nFor $ 196 = 2^2 \\times 7^2 $, the exponents are 2 and 2.\n\nSo the number of positive divisors is:\n\n$$\n(2 + 1)(2 + 1) = 3 \\times 3 = 9\n$$\n\n---\n\n### **Final Answer**\n\nTherefore, the number of positive whole-number divisors of 196 is:\n\n$$\n\\boxed{9}\n$$\n\nTherefore, the final answer is: $\\boxed{9}$. I hope it is correct.`"

---

> ### Author Response · Authors · 2025-11-26
> **We really look forward to further discussion**
>
> Dear Reviewer EUV3,
>
> We genuinely thank you again for your time & efforts and your constructive comments. We really look forward to discussing further with you to see if our response answers your questions.
>
> We wish reviewer EUV3 could kindly check our response. Thank you!

---

> > ### Comment · Reviewer_EUV3 · 2025-11-26
> >
> > Thanks for the authors' detailed response. The concerns are mostly addressed, and I raise my ratings.

---

> > > ### Author Response · Authors · 2025-11-26
> > >
> > > Thank Reviewer EUV3 so much for your constructive questions and feedback to help us improve our work! We truly appreciate your time and effort spent on reviewing our paper and raising your ratings!

---

### Official Review · Reviewer_zM6i · 2025-11-01

**Soundness:** 2
**Presentation:** 3
**Contribution:** 3
**Rating:** 6
**Confidence:** 3

**Summary:**

The paper proposes Sonata, a lightweight adapter that predicts whether and how much chain-of-thought “thinking” an LLM should use by estimating self-consistency directly from the last-layer hidden state during prefilling. Using predicted self-consistency as a proxy for reasoning difficulty, Sonata allocates thinking budgets via a simple threshold, enabling non-thinking for easy queries and CoT for hard ones with negligible overhead. Across multiple models (8B–235B) and benchmarks (AIME25, GSM8K, MATH-500, GPQA), it reduces thinking tokens by 20–60% at the same accuracy, or improves accuracy by up to 2% at the same cost, while also lowering latency. The adapter is task-agnostic once trained and is compatible with existing CoT compression/pruning methods.

**Strengths:**

- Minimal overhead with substantial efficiency gains: Adds virtually zero compute (<0.1% layer-equivalent), reduces thinking tokens by 20–60%, and cuts end-to-end latency by ~27–36% across models.

- Better performance–efficiency trade-off: Self-consistency–guided allocation outperforms fixed budgets and self-judging, maintaining accuracy or improving it (up to +2%) at the same cost.

- Generalizable and compatible: Once trained per model, the adapter transfers across tasks (e.g., math to GPQA) and integrates seamlessly with existing CoT compression/pruning and early-exit strategies.

**Weaknesses:**

- Missing strong baselines: Lacks systematic comparisons against recent dynamic budget/early-stop/CoT-pruning methods, so relative advantages are not fully established.

- Narrow task coverage: Evaluation focuses on math/scientific reasoning; absent tests on code generation, open-ended generation, and  long-context multi-step planning

- Coarse and potentially brittle control: Online decision is primarily binary via a single threshold

**Questions:**

How sensitive is Sonata to offline calibration choices (calibration set size and difficulty distribution, number of samples N per query, and verifier noise), and can similar performance be maintained under lower calibration cost?

---

> ### Author Response · Authors · 2025-11-22
> **Response to Reviewer zM6i (1/3)**
>
> Dear Reviewer zM6i,
>
> Thank you for your positive feedback on our "minimal overhead with substantial efficiency gains" and being "generalizable and compatible". We sincerely appreciate your recognition of our effort. Below, we provide point-to-point responses to address your questions.

---

> ### Author Response · Authors · 2025-11-22
> **Response to Reviewer zM6i (2/3)**
>
> ### [W1. Comparisons/Integration with recent methods.]
>
> Thanks for the suggestions. Our method is "generalizable and compatible" with all of these recent "dynamic budget/early-stop/CoT-pruning" baselines, as illustrated in Section 1 (around line 70) of our original draft and mentioned by Reviewer zM6i/SHJi/EUV3; therefore, our method can be integrated with any of them to further improve efficiency.
>
> To better answer your question, we conduct experiments by combining our Sonata with REFRAIN **[1]** (an early-stop method for thinking). Specifically, we first apply our Sonata adapter to decide whether to think or not; if thinking, then REFRAIN is applied during thinking to determine when to stop it. As shown by the results in **Table R1**, integrating our Sonata (adaptive thinking) with REFRAIN (early-stop thinking) together further improves the efficiency by reducing 15% number of tokens on average while maintaining the performance. This validates that our Sonata is compatible with existing dynamic budget/early-stop/CoT-pruning for further efficiency. We include these additional results in our revised draft.
>
> **[Table R1] (Accuracy; # of tokens)**
>
> |                                  | AIME25             | MATH-500         | GSM8K             | GPQA             | Average          |
> | -------------------------------- | ------------------ | ---------------- | ----------------- | ---------------- | ---------------- |
> | Qwen3-8B                         | 60.0; 16995        | 97.6; 4900       | 95.2; 1994        | 60.1; 7458       | 78.2; 7837       |
> | w. Const Budget                  | 30.0; 4096 (24%)   | 93.2; 4096 (84%) | 95.0; 4096 (205%) | 57.1; 4096 (55%) | 68.8; 4096 (52%) |
> | w. Self-Judge | 60.0; 17019 (100%) | 96.0; 4315 (88%) | 92.7; 1076 (54%)  | 57.6; 5913 (79%) | 76.6; 7080 (90%) |
> | w. Sonata                        | 63.3; 16449 (97%)  | 97.4; 3694 (75%) | 95.6; 890 (45%)   | 62.0; 3590 (48%) | 79.6; 6156 (79%) |
> | w. Sonata + REFRAIN              | 60.0; 12840 (76%)  | 97.6; 3309 (68%) | 95.0; 845 (42%)   | 61.8; 2958 (40%) | 78.7; 4988 (64%) |
>
> ### [W2. Narrow task coverage.]
>
> Thanks for the suggestions. As you mentioned, our Sonata is "generalizable" across tasks. To further validate this and better address your concerns. We further conduct comparison experiments across baselines on the LiveCodeBench task (code generation) with Qwen3-8B and Qwen3-32B models. As shown by the LiveCodeBench results in **Table R2** and **R3**, Sonata still consistently outperforms all other baselines, with highest pass@1 across all methods and reduced numbers of tokens compared to the vanilla models. We further include this new code generation task in our revised draft.
>
> **[Table R2] (Accuracy; # of tokens)**
>
> |                 | AIME25             | MATH-500         | GSM8K             | GPQA             | LiveCodeBench      | Average           |
> | --------------- | ------------------ | ---------------- | ----------------- | ---------------- | ------------------ | ----------------- |
> | Qwen3-8B        | 60.0; 16995        | 97.6; 4900       | 95.2; 1994        | 60.1; 7458       | 57.8; 14421        | 74.1; 9154        |
> | w. Const Budget | 30.0; 4096 (24%)   | 93.2; 4096 (84%) | 95.0; 4096 (205%) | 57.1; 4096 (55%) | 49.3; 4096 (28%)   | 64.9; 4096 (45%)  |
> | w. Self-Judge   | 60.0; 17019 (100%) | 96.0; 4315 (88%) | 92.7; 1076 (54%)  | 57.6; 5913 (79%) | 57.1%; 13496 (93%) | 72.7%; 8364 (91%) |
> | w. Sonata       | 63.3; 16449 (97%)  | 97.4; 3694 (75%) | 95.6; 890 (45%)   | 62.0; 3590 (48%) | 58.2; 13054 (90%)  | 75.3; 7535 (82%)  |
>
> **[Table R3] (Accuracy; # of tokens)**
>
> |                 | AIME25             | MATH-500          | GSM8K             | GPQA             | LiveCodeBench     | Average          |
> | --------------- | ------------------ | ----------------- | ----------------- | ---------------- | ----------------- | ---------------- |
> | Qwen3-32B       | 70.0; 14971        | 97.6; 3670        | 94.8; 1654        | 63.6; 5329       | 63.8; 13729       | 78.0; 7853       |
> | w. Const Budget | 56.7; 4096 (27%)   | 95.8; 4096 (112%) | 95.3; 4096 (248%) | 64.1; 4096 (77%) | 47.0; 4096 (30%)  | 69.8; 4096 (52%) |
> | w. Self-Judge   | 73.3; 15625 (104%) | 96.2; 3107 (85%)  | 93.1; 666 (40%)   | 54.5; 3807 (71%) | 59.3; 13360 (97%) | 75.3; 7313 (93%) |
> | w. Sonata       | 70.0; 14890 (100%) | 98.0; 2583 (70%)  | 94.4; 728 (44%)   | 63.1; 3568 (70%) | 63.4; 13600 (99%) | 77.8; 7074 (90%) |
>
> **[1]** Sun, R., Cheng, W., Li, D., Chen, H. and Wang, W., 2025. Stop When Enough: Adaptive Early-Stopping for Chain-of-Thought Reasoning. arXiv preprint arXiv:2510.10103.

---

> ### Author Response · Authors · 2025-11-22
> **Response to Reviewer zM6i (3/3)**
>
> ### [W3. Coarse and potentially brittle control.]
>
> Thanks for the suggestion. We conduct additional ablation studies with more fine-grained control. Specifically, following GPT-OSS, Gemini 3, etc., we extend current binary thinking decision of Sonata to 4 modes of {non-thinking (0 thinking tokens), low thinking (<1024 thinking tokens), medium thinking (<4096 thinking tokens), high thinking (any number of thinking tokens)}, with adapter thresholds of {0.3, 0.5, 0.7, 0.9} respectively. We evaluate it on the Qwen3-8B model across all tasks for comparison.
>
> As shown by the results in **Table R4**, Sonata with fine-grained thinking control seems to be more brittle than our original Sonata with binary thinking decision, potentially due to the complex threshold combinations that are difficult to tune. Future work could explore data-driven or training-based methods to find optimal thresholds of the fine-grained thinking control. We further include these additional ablation results and discussions in our revised draft.
>
> **[Table R4] (Accuracy; # of tokens)**
>
> |                                    | AIME25             | MATH-500         | GSM8K             | GPQA             | Average          |
> | ---------------------------------- | ------------------ | ---------------- | ----------------- | ---------------- | ---------------- |
> | Qwen3-8B                           | 60.0; 16995        | 97.6; 4900       | 95.2; 1994        | 60.1; 7458       | 78.2; 7837       |
> | w. Const Budget                    | 30.0; 4096 (24%)   | 93.2; 4096 (84%) | 95.0; 4096 (205%) | 57.1; 4096 (55%) | 68.8; 4096 (52%) |
> | w. Self-Judge                      | 60.0; 17019 (100%) | 96.0; 4315 (88%) | 92.7; 1076 (54%)  | 57.6; 5913 (79%) | 76.6; 7080 (90%) |
> | w. Sonata (binary, original)       | 63.3; 16449 (97%)  | 97.4; 3694 (75%) | 95.6; 890 (45%)   | 62.0; 3590 (48%) | 79.6; 6156 (79%) |
> | w. Sonata (fine-grained, 4 levels) | 63.3; 16449 (97%)  | 96.2; 3258 (66%) | 95.4; 853 (43%)   | 58.8; 2983 (40%) | 78.4; 5886 (75%) |
>
> ### [Q1. Calibration dataset size and lower calibration cost.]
>
> Thanks for your questions. We conduct further ablation studies to address your concerns. Specifically, we evaluate Sonata with two randomly sampled subsets of 100 and 200 samples, respectively, on Qwen3-8B with AIME25, MATH-500, GSM8K, and GPQA for comparison. As shown in **Table R5**, Sonata with the calibration dataset size of 1000, 200 samples, or even 100 samples, consistently outperforms other baselines, in terms of accuracy and efficiency (number of tokens). We further include these additional ablation in our revised draft.
>
> **[Table R5] (Accuracy; # of tokens)**
>
> |                                  | AIME25             | MATH-500         | GSM8K             | GPQA             | Average           |
> | -------------------------------- | ------------------ | ---------------- | ----------------- | ---------------- | ----------------- |
> | Qwen3-8B                         | 60.0; 16995        | 97.6; 4900       | 95.2; 1994        | 60.1; 7458       | 78.2; 7837        |
> | w. Const Budget                  | 30.0; 4096 (24%)   | 93.2; 4096 (84%) | 95.0; 4096 (205%) | 57.1; 4096 (55%) | 68.8; 4096 (52%)  |
> | w. Self-Judge                    | 60.0; 17019 (100%) | 96.0; 4315 (88%) | 92.7; 1076 (54%)  | 57.6; 5913 (79%) | 76.6; 7080 (90%)  |
> | w. Sonata (1k samples, original) | 63.3; 16449 (97%)  | 97.4; 3694 (75%) | 95.6; 890 (45%)   | 62.0; 3590 (48%) | 79.6; 6156 (79%)  |
> | w. Sonata (200 samples)          | 60.0; 16990 (100%) | 97.0; 3750 (77%) | 95.4; 865 (43%)   | 61.3; 3483 (47%) | 78.4; 6272 (80%)  |
> | w. Sonata (100 samples)          | 63.3; 17005 (100%) | 96.8; 3883 (79%) | 95.0; 1005 (50%)  | 60.9; 3994 (54%) | 79.0%; 6472 (83%) |

---

> ### Author Response · Authors · 2025-11-26
> **We really look forward to further discussion**
>
> Dear Reviewer zM6i,
>
> We thank you again for the time and effort on reviewing our work and your constructive comments. We really hope to discuss further with you to see if our response answers your questions.
>
> We genuinely hope reviewer zM6i could kindly check our response. Thank you so much!

---

### Official Review · Reviewer_SHJi · 2025-11-11

**Soundness:** 3
**Presentation:** 3
**Contribution:** 2
**Rating:** 6
**Confidence:** 4

**Summary:**

This paper addresses the critical challenge of compute-optimal inference for large language models (LLMs) with chain-of-thought (CoT) reasoning capabilities. A key limitation of existing methods is the lack of adaptive thinking budget allocation: fixed budgets either waste resources on simple queries or lead to errors on complex ones, while entropy-based proxies fail to capture intrinsic reasoning difficulty. The authors propose two core insights: (1) self-consistency (agreement among multiple reasoning paths for a query) is a reliable proxy for thinking necessity—low self-consistency indicates a need for extended reasoning, while high self-consistency signals minimal thinking is required; (2) self-consistency patterns are highly distinguishable in the LLM’s latent space (especially in deeper layers), enabling efficient prediction without expensive repeated sampling.

Building on these insights, the authors introduce Sonata (Self-Consistency-Guided Adapter for Thinking Allocation): a lightweight 2-layer MLP adapter trained offline on a calibration dataset to predict self-consistency directly from the last-layer hidden representations of queries during the prefilling stage. At inference, Sonata uses this prediction to dynamically allocate thinking budgets (e.g., skipping thinking for high self-consistency queries) with <1‰ computational overhead. Extensive experiments across 4 models (Qwen3-8B to Qwen3-235B-A22B) and 4 benchmarks (AIME25, GSM8K, MATH500, GPQA) show Sonata reduces thinking tokens by 20–60% while maintaining or improving accuracy (up to 2% gain). It also generalizes across tasks and is compatible with existing CoT compression methods.

**Strengths:**

- Experiments are comprehensive, covering diverse model sizes (8B to 235B parameters) and task types (mathematical, general reasoning), ensuring results are not limited to specific settings.

- Unlike black-box adaptive methods, Sonata’s reliance on self-consistency and latent space patterns provides clear interpretability—users can trace budget decisions to a measurable proxy (self-consistency) rather than opaque model outputs.

- Sonata’s minimal overhead and offline training make it easy to integrate into existing LLM pipelines. Compatibility with CoT compression enables further efficiency gains, enhancing real-world deployment value.

**Weaknesses:**

- Sonata requires a calibration dataset (1000 samples from OpenMathReasoning) for offline training. The paper does not explore how calibration set size or difficulty affects performance—e.g., would smaller datasets or datasets with mismatched difficulty (e.g., training on easy queries, testing on hard ones) degrade Sonata’s accuracy? This is critical for low-resource scenarios.

- The paper mentions a cluster of “intrinsically difficult queries” (lower-left of Figure 2) where even extended reasoning fails, but does not address how Sonata handles these. For example, could Sonata be modified to detect such queries and avoid wasting tokens on futile reasoning?

- The self-judge baseline uses a binary “YES/NO” prompt for thinking need, but more nuanced prompts (e.g., “rate difficulty 1–5”) might improve its performance. The paper’s current self-judge design could underestimate the potential of LLM self-assessment, weakening the comparison to Sonata.

**Questions:**

1. How does Sonata’s performance degrade when the calibration dataset is smaller (e.g., 100 samples) or has difficulty distributions mismatched to the test set (e.g., calibration on easy queries, test on hard queries)? For real-world use, users may not have access to large, well-matched calibration data—insights here would improve practicality.

2. For the “intrinsically difficult queries” (low self-consistency, low thinking gain), does Sonata still allocate thinking budgets? If so, could you extend Sonata to predict both self-consistency and thinking gain (e.g., via a multi-output adapter) to avoid wasting tokens on these queries?

3. The self-judge baseline uses a binary prompt, but prior work shows nuanced prompts improve LLM self-assessment. Have you tested Sonata against a self-judge baseline with difficulty-rating prompts (e.g., “rate how much thinking this query needs: 1–5”)?

---

> ### Author Response · Authors · 2025-11-22
> **Response to Reviewer SHJi (1/3)**
>
> Dear Reviewer SHJi,
>
> We thank you for the positive feedback on the "comprehensive experiments", "clear interpretability", and "Compatibility with CoT compression". Your recognition of our effort is deeply appreciated. To further answer your questions, we provide point-to-point responses as follows.

---

> ### Author Response · Authors · 2025-11-22
> **Response to Reviewer SHJi (2/3)**
>
> ### [W1. Different calibration set size.]
>
> Thanks for your questions. We conduct further ablation studies to address your concerns. Specifically, we evaluate Sonata with two randomly sampled subsets of 100 and 200 samples respectively, on Qwen3-8B with AIME25, MATH-500, GSM8K, and GPQA for comparison. As shown in **Table R1**, Sonata with the calibration dataset size of 1000, 200, or even 100 samples, consistently outperforms other baselines, in terms of accuracy and efficiency (number of tokens). We further include these additional ablation studies in our revised draft.
>
> **[Table R1] (Accuracy; # of tokens)**
>
> |                                  | AIME25             | MATH-500         | GSM8K             | GPQA             | Average           |
> | -------------------------------- | ------------------ | ---------------- | ----------------- | ---------------- | ----------------- |
> | Qwen3-8B                         | 60.0; 16995        | 97.6; 4900       | 95.2; 1994        | 60.1; 7458       | 78.2; 7837        |
> | w. Const Budget                  | 30.0; 4096 (24%)   | 93.2; 4096 (84%) | 95.0; 4096 (205%) | 57.1; 4096 (55%) | 68.8; 4096 (52%)  |
> | w. Self-Judge                    | 60.0; 17019 (100%) | 96.0; 4315 (88%) | 92.7; 1076 (54%)  | 57.6; 5913 (79%) | 76.6; 7080 (90%)  |
> | w. Sonata (1k samples, original) | 63.3; 16449 (97%)  | 97.4; 3694 (75%) | 95.6; 890 (45%)   | 62.0; 3590 (48%) | 79.6; 6156 (79%)  |
> | w. Sonata (200 samples)          | 60.0; 16990 (100%) | 97.0; 3750 (77%) | 95.4; 865 (43%)   | 61.3; 3483 (47%) | 78.4; 6272 (80%)  |
> | w. Sonata (100 samples)          | 63.3; 17005 (100%) | 96.8; 3883 (79%) | 95.0; 1005 (50%)  | 60.9; 3994 (54%) | 79.0%; 6472 (83%) |
>
> ### [W2. Handling intrinsically difficult queries.]
>
> Thanks for your kind suggestion. In our experiments, these "intrinsically difficult queries" (as shown on the lower-left of Figure 2) are rare. To further validate this, we extend Sonata by training two adapters to predict self-consistency and thinking gain, respectively. The thinking decision is adapted as: Only if the predicted self-consistency < 0.3 and the predicted thinking gain > 0.1, the model will conduct thinking; otherwise, the model proceeds without thinking. It is evaluated on Qwen3-8B with AIME25, MATH-500, GSM8K, and GPQA. As shown in **Table R2**, the extended Sonata, which predicts both self-consistency and thinking gain, performs nearly the same as the original Sonata, which predicts only self-consistency, validating our observations that "intrinsically difficult queries" are rare and have negligible impact on the accuracy and efficiency. We further include these extra results in our revised draft.
>
> **[Table R2] (Accuracy; # of tokens)**
>
> |                                            | AIME25             | MATH-500         | GSM8K             | GPQA             | Average          |
> | ------------------------------------------ | ------------------ | ---------------- | ----------------- | ---------------- | ---------------- |
> | Qwen3-8B                                   | 60.0; 16995        | 97.6; 4900       | 95.2; 1994        | 60.1; 7458       | 78.2; 7837       |
> | w. Const Budget                            | 30.0; 4096 (24%)   | 93.2; 4096 (84%) | 95.0; 4096 (205%) | 57.1; 4096 (55%) | 68.8; 4096 (52%) |
> | w. Self-Judge                              | 60.0; 17019 (100%) | 96.0; 4315 (88%) | 92.7; 1076 (54%)  | 57.6; 5913 (79%) | 76.6; 7080 (90%) |
> | w. Sonata (self-consistency, original)     | 63.3; 16449 (97%)  | 97.4; 3694 (75%) | 95.6; 890 (45%)   | 62.0; 3590 (48%) | 79.6; 6156 (79%) |
> | w. Sonata (self-consistency+thinking gain) | 63.3; 16985 (100%) | 97.2; 3650 (74%) | 95.0; 905 (45%)   | 61.7; 3483 (47%) | 79.3; 6256 (80%) |

---

> ### Author Response · Authors · 2025-11-22
> **Response to Reviewer SHJi (3/3)**
>
> ### [W3. Fine-grained self-judge baseline.]
>
> Thanks for the question. We conduct additional experiments with "rate difficulty 1–5" self-judge baseline on Qwen3-8B model with AIME25, MATH-500, GSM8K, and GPQA for comparison. Specifically, the model will first read the query and predict a difficulty S ∈ {1, 2, 3, 4, 5}, and then conduct thinking with (S-1) * 2048 tokens of budget (S=1 refers to no thinking).
>
> As shown in **Table R3**, the fine-grained self-judge baseline (5-level) seems to hurt the performance compared to the binary self-judge, e.g., 94.6% vs. 96.0% accuracy on MATH-500 with a similar number of tokens (4323 vs. 4315). This is potentially because it's hard for the models to decide a more fine-grained thinking budget without any training or adaptation, while our Sonata consistently outperforms both binary and 5-level self-judge baselines in terms of both performance and efficiency. Moreover, self-judge baselines require one additional model generation pass to conduct the judge, which leads to more computing cost that is not reflected in our used metrics (ie, the number of tokens to generate thinking CoT and final answer). We further include these additional results and discussions in our revised draft.
>
> **[Table R3] (Accuracy; # of tokens)**
>
> |                                  | AIME25             | MATH-500         | GSM8K             | GPQA             | Average          |
> | -------------------------------- | ------------------ | ---------------- | ----------------- | ---------------- | ---------------- |
> | Qwen3-8B                         | 60.0; 16995        | 97.6; 4900       | 95.2; 1994        | 60.1; 7458       | 78.2; 7837       |
> | w. Const Budget                  | 30.0; 4096 (24%)   | 93.2; 4096 (84%) | 95.0; 4096 (205%) | 57.1; 4096 (55%) | 68.8; 4096 (52%) |
> | w. Self-Judge (binary, original) | 60.0; 17019 (100%) | 96.0; 4315 (88%) | 92.7; 1076 (54%)  | 57.6; 5913 (79%) | 76.6; 7080 (90%) |
> | w. Self-Judge (5-level)          | 60.0; 16990 (100%) | 94.6; 4323 (88%) | 94.3; 2350 (118%) | 56.1; 5857 (79%) | 76.3; 7380 (94%) |
> | w. Sonata                        | 63.3; 16449 (97%)  | 97.4; 3694 (75%) | 95.6; 890 (45%)   | 62.0; 3590 (48%) | 79.6; 6156 (79%) |
>
> ### [Q1. Different calibration set size.]
>
> Please kindly see our response to W1.
>
> ### [Q2. Intrinsically difficult queries.]
>
> Please kindly see our response to W2.
>
> ### [Q3. Fine-grained self-judge baseline.]
>
> Please kindly see our response to W3.

---

> ### Author Response · Authors · 2025-11-26
> **We really look forward to further discussion**
>
> Dear Reviewer SHJi,
>
> We thank reviewer SHJi for the time on reviewing and the constructive feedback again. We really hope to discuss further with you to see if our response answers your questions.
>
> We genuinely hope reviewer SHJi could kindly check our response. Thank you very much!

---

### Author Response · Authors · 2025-11-23
**General Response**

Dear PCs, SACs, ACs, and Reviewers,

We are sincerely grateful for the time and effort that everyone has devoted to evaluating our paper. We also particularly appreciate that all reviews appear to be carefully written by humans, given that some other submissions at this venue seem to have suffered from LLM-generated reviews. We are thankful for the reviewers' positive acknowledgment of our work as **"novel and well supported by extensive experiments," with "clear interpretability," "substantial efficiency gains," and being "well-written and easy to follow," as well as "simple and practical."**

We also genuinely appreciate the constructive feedback and suggestions that were provided. Below, in addition to our detailed point-by-point responses, we summarize the key updates made to the paper.

### [Additional experiments]

-   **[Calibration set size ablation]** As suggested by Reviewers **SHJi** and **zM6i**, we evaluate Sonata with smaller calibration datasets (100 and 200 samples) on Qwen3-8B across all benchmarks. Results show consistent performance even with 100 samples, validating robustness under low-resource scenarios, presented in Section 5.2 of our revised draft.
-   **[Integration with existing methods]** As mentioned by Reviewers **zM6i** and **sPcD**, we conduct experiments combining Sonata with REFRAIN (an early-stop method). Results demonstrate 15% additional token reduction on average while maintaining accuracy, validating compatibility with existing dynamic budget/early-stop methods, presented in Section 5.2 of our revised draft.
-   **[Extended task coverage]** As suggested by Reviewers **zM6i** and **EUV3**, we evaluate Sonata on LiveCodeBench (code generation task) with Qwen3-8B and Qwen3-32B. Results show consistent improvements, together with existing GPQA, further validating cross-domain generalization beyond math calibration data, presented in Section 5.1 of our revised draft.
-   **[Extended Sonata with thinking gain prediction]** As suggested by Reviewer **SHJi**, we train an extended adapter predicting both self-consistency and thinking gain to handle "intrinsically difficult queries." Results show negligible impact, confirming such queries are rare, presented in Appendix C.1 of our revised draft.
-   **[Fine-grained Self-Judge baseline]** As suggested by Reviewer **SHJi**, we evaluate a 5-level difficulty-rating self-judge baseline. Results show it underperforms the binary version, validating our baseline design, presented in Appendix C.2 of our revised draft.
-   **[Fine-grained thinking control]** As suggested by Reviewer **zM6i**, we extend Sonata to 4-level thinking control (non/low/medium/high thinking). Results indicate increased brittleness due to complex threshold tuning, confirming our binary design choice, presented in Appendix C.3 of our revised draft.
-   **[Multi-position and multi-layer ablation]** As suggested by Reviewer **sPcD**, we evaluate Sonata with concatenated hidden states from multiple positions/layers. Results confirm our design of using the last token from the last layer achieves the best performance, echoing our presented motivation in Figure 3 of our draft, presented in Section 5.2 of our revised draft.

### [Paper edit]

-   All additional experiment results and corresponding discussions that are listed above.
-   Extended Figure 3 with PCA visualization on GPQA (scientific reasoning task), validating that self-consistency is distinguishable across domains. (@Reviewer **EUV3**)
-   Justification for decision threshold τ₀=0.3 with concrete examples comparing questions at different difficulty levels around this threshold, presented in Appendix C.4 of our revised draft. (@Reviewer **EUV3**)
-   Concrete examples comparing model outputs before and after applying Sonata, illustrating the difference between thinking and non-thinking modes, presented in Appendix C.5 of our revised draft.  (@Reviewer **EUV3**)
-   Discussion on potential extension to dynamic reasoning length decision during long-form reasoning as future work, presented in Appendix D of our revised draft.  (@Reviewer **sPcD**)

We hope our point-to-point responses below will clarify any remaining questions from the reviewers. Please feel free to let us know if there are any further questions. We will be more than happy to address them.

We extend our sincere appreciation once again for the reviewers' time, effort, and human-written feedback.

Warmest regards,

Authors of Submission #975

---

### Meta-Review · Area_Chair_t8kk · 2026-01-04

**Summary:**

The paper introduces "Sonata," a lightweight adapter designed to predict the necessity of Chain-of-Thought (CoT) reasoning during the prefilling stage of Large Language Models (LLMs). By using self-consistency as a proxy for reasoning difficulty, Sonata employs a simple binary gating mechanism (think vs. do not think) based on the last-layer hidden states. The method is trained offline on a calibration set and incurs negligible computational overhead (<0.1%). Empirical results across models ranging from 8B to 235B parameters demonstrate significant efficiency gains (20–60% token reduction) while maintaining or improving accuracy on benchmarks like MATH, GSM8K, and GPQA.

**Reviewer Concerns:**

While the authors did not explicitly test "difficulty distribution mismatch" as requested by Reviewer SHJi , the successful zero-shot transfer from Math to Science and Code domains provides strong implicit evidence of the method's robustness to distribution shifts. Additionally, while a direct head-to-head comparison with only REFRAIN was not provided, the integration results sufficiently demonstrate the value of Sonata as a "gatekeeper" module.

**Reviewer Scores:**

As mentioned by the authors, Reviewer sPcD increased the score from 4 to 6 on November 23, and Reviewer EUV3 increased the score from 4 to 6 on November 26, both of which occurred well before the November 27 data leak incident.

I think the final scores of this paper are 6 6 6 6.

---

### Decision · Program_Chairs · 2026-01-26

Accept (Poster)